# Clinically used broad-spectrum antibiotics compromise inflammatory monocyte-dependent antibacterial defense in the lung

Patrick J. Dörner[1], Harithaa Anandakumar[2,3,4,5], Ivo Röwekamp [1], Facundo Fiocca Vernengo [1], Belén Millet Pascual-Leone [1], Marta Krzanowski [1], Josua Sellmaier [1], Ulrike Brüning [6], Raphaela Fritsche-Guenther[6], Lennart Pfannkuch[1], Florian Kurth [1], Miha Milek [7], Vanessa Igbokwe[1], Ulrike Löber [2,3,4], Birgitt Gutbier[1], Markus Holstein[1], Gitta Anne Heinz [8], Mir-Farzin Mashreghi [8], Leon N. Schulte[9,10], Ann-Brit Klatt[1], Sandra Caesar[1], Sandra-Maria Wienhold [1], Stefan Offermanns [11], Matthias Mack [12], Martin Witzenrath[1,13], Stefan Jordan [14], Dieter Beule[7], Jennifer A. Kirwan[6], Sofia K. Forslund [2,3,4,15], Nicola Wilck [2,3,4,5], Hendrik Bartolomaeus [2,3,4,5], Markus M. Heimesaat[14] & Bastian Opitz [1,13] ✉

Hospital-acquired pneumonia (HAP) is associated with high mortality and costs, and frequently caused by multidrug-resistant (MDR) bacteria. Although prior antimicrobial therapy is a major risk factor for HAP, the underlying mechanism remains incompletely understood. Here, we demonstrate that antibiotic therapy in hospitalized patients is associated with decreased diversity of the gut microbiome and depletion of short-chain fatty acid (SCFA) producers. Infection experiments with mice transplanted with patient fecal material reveal that these antibiotic-induced microbiota perturbations impair pulmonary defense against MDR *Klebsiella pneumoniae*. This is dependent on inflammatory monocytes (IMs), whose fatty acid receptor (FFAR)2/3-controlled and phagolysosome-dependent antibacterial activity is compromised in mice transplanted with antibiotic-associated patient microbiota. Collectively, we characterize how clinically relevant antibiotics affect antimicrobial defense in the context of human microbiota, and reveal a critical impairment of IM´s antimicrobial activity. Our study provides additional arguments for the rational use of antibiotics and offers mechanistic insights for the development of novel prophylactic strategies to protect high-risk patients from HAP.

Hospital-acquired pneumonia (HAP), including ventilator-associated pneumonia, is a frequent complication in hospital care, and the leading cause of death from hospital-acquired infections[1]. The European Centre for Disease Prevention and Control estimates a burden of more than 700,000 annual cases of HAP in European countries alone[2]. They dramatically increase both length of hospital stay and healthcare costs, and are associated with an attributable mortality of 4 to 13%[2,3]. HAPs are frequently caused by multidrug-resistant (MDR) bacteria such as carbapenemase-producing *Klebsiella pneumoniae*. Prior antimicrobial therapy is a well-known risk factor for HAP[4,

HAP-associated death[5,6] and HAP caused by MDR organisms[7,8]. A recently published retrospective cohort study of over 3000 critically ill patients indicated that specifically antibiotics with activity against anaerobic bacteria increase the risk for HAP and other nosocomial infections caused by Enterobacteriaceae[9]. However, our knowledge of how antimicrobial therapy affects susceptibility to HAP is still incomplete, and no targeted prophylactic interventions exist to protect patients receiving antimicrobial therapy from subsequent HAP.

Asymptomatic intestinal and/or oropharyngeal colonization is considered an essential step in *K. pneumoniae* pathogenesis[10], and *K. pneumoniae* colonization is normally suppressed by healthy microbiota through a mechanism known as colonization resistance[11–13]. Antimicrobial therapy, however, can disrupt the microbiota as an adverse side effect, thereby compromising colonization resistance and allowing overgrowth of, for example, MDR *K. pneumoniae*. In addition, the microbiota primes and calibrates immune defense pathways to protect against pulmonary infection[14–17]. These mechanisms have been found to depend on microbiota-derived metabolites and other microbial molecules, and have so far mainly been characterized by comparing germ-free or microbiota-depleted mice with conventionally colonized mice[18–22]. In contrast, comparatively little is known about the impact of clinical antimicrobial therapies on the human microbiota and how this affects antimicrobial defense mechanisms.

Here, we hypothesize that antimicrobial therapy perturbs the gut microbiota in hospitalized patients and that these microbiota alterations influence immune defense against MDR *K. pneumoniae* in patient microbiota-transplanted mice. We reveal that microbiota perturbations induced by clinically used broad-spectrum antibiotics enhance susceptibility to pulmonary infection by reducing FFAR2/3-controlled antibacterial activity of inflammatory monocytes (IMs).

## Results

### Antimicrobial therapy in patients is associated with reduced diversity of the gut microbiota and decreased abundance of SCFA producers

To characterize the effect of widely used broad-spectrum antibiotics on the gut microbiota of hospitalized patients and subsequently test their influence on pulmonary immune defense in fecal microbiome transplanted animals, we collected fecal samples from 72 hospitalized patients receiving antibiotics or not. All patients in the antibiotic group were given parenterally a commonly in the hospital setting used broad-spectrum beta-lactam (in most cases either a carbapenem or piperacillin-tazobactam; one patient received ampicillin-sulbactam) for at least two days (patients characteristics are shown in Table 1 and supplementary data 1). Cultural analysis of a subgroup of these samples suggests that the microbiota of patients receiving antibiotics and patients without antibiotics did not differ in terms of total bacterial loads (Fig. 1A). Shotgun metagenomic sequencing was conducted on stool samples from 55 patients, including 26 patients receiving broad-spectrum antibiotic therapy and 29 without antimicrobial therapy. While there was no difference on phylum level (Fig. S1), the gut microbiota from patients receiving antibiotics exhibited significantly decreased taxonomic diversity and were characterized by lower abundances of several commensal species belonging to the order Clostridiales such as *Ruminococcus torques*, *Faecalibacterium prausnitzii*, and *Dorea longicatena* when compared to microbiota of patients without antimicrobial therapy (Fig. 1B–D). Many of these gut microbes are known to produce short-chain fatty acids (SCFA)[23], and functional profiling indicated depletion of functional modules linked to SCFA production in the antibiotic-associated gut microbiota (Fig. 1E). Analysis of SCFA levels in plasma samples of a subgroup of patients showed that the production of several SCFAs was impaired in individuals receiving antibiotics (Fig. 1F–J). Overall, our data indicate that broad-spectrum antimicrobial therapy is associated with compositional changes of the gut microbiota and their reduced capacity to produce SCFAs.

### Human microbiota transplantation experiments reveal that antibiotic-induced microbiota alterations influence antibacterial immunity in the lung via SCFA receptors

Next, we aimed to characterize the effects of broad-spectrum antibiotics on antibacterial defenses in the lung in the context of the human microbiota and to test for possible involvement of SCFA receptors. Therefore, wild-type (WT) and *Ffar2/Ffar3*[-/-] mice (lacking the SCFA receptors FFAR2 and FFAR3) were depleted of their own microbiota with an antibiotic cocktail, subsequently transplanted with fecal material from our patients either receiving antibiotics (antibiotic-associated microbiota) or not receiving antibiotics (antibiotic-naïve microbiota) (Fig. 2A). Twenty-six different patient stool samples were used to transplant 46 animals, so that in most cases pairs of WT and *Ffar2/Ffar3*[-/-] mice (and in a few cases individual mice alone) received an individual patient microbiota. Metagenomic sequencing of fecal samples from recipient mice and comparison with the donor material indicated that the microbiota and relevant functional changes induced by antibiotics, e.g. the depletion of anaerobic commensals and functional modules linked to SCFA production, were successfully transferred (Fig. 2B, C, Fig. S2A, B). Microbiota-transplanted mice were then intranasally infected with a clinical isolate of a carbapenemase-producing MDR *K. pneumoniae* (belonging to the epidemic sequence type (ST) 258) to assess pathogen loads. 24 h post infection, WT mice transplanted with antibiotic-naïve microbiota had lower bacterial loads in the lungs as compared to WT mice transplanted with antibiotic-associated microbiota (Fig. 2D). Likewise, *Ffar2/Ffar3*[-/-] mice displayed an impaired bacterial control. Of note, no significant difference was observed between *Ffar2/Ffar3*[-/-] receiving antibiotic-associated microbiota or antibiotic-naïve microbiota, with the fold-changes related to antibiotic therapy being significantly higher in WT as compared to *Ffar2/Ffar3*[-/-] mice (Fig. 2E). However, the slight trending difference in bacterial loads between *Ffar2/Ffar3*[-/-] mice receiving antibiotic-associated or antibiotic-naïve microbiota suggests that FFAR2/FFAR3-indepenent effects of the antibiotic-induced microbiota disturbance on antibacterial resistance may exist. Moreover, we observed higher levels of myeloperoxidase (MPO) and serum albumin leakage into bronchoalveolar lavage fluids (BALF) of WT mice transplanted with antibiotic-associated microbiota as compared to BALF of mice transplanted with antibiotic-naïve microbiota (Fig. 2F, G), indicating enhanced neutrophilic inflammation and lung barrier damage.

### SCFA receptor deficiency has little influence on pulmonary cytokine production and immune cell infiltration during *K. pneumoniae* ST258 infection

To define mechanistically how SCFAs affect antibacterial immunity against lung infections with *K. pneumoniae* ST258, we used conventionally colonized SPF and microbiota-depleted WT and *Ffar2/Ffar3*[-/-] mice. Similar to human microbiota-transplanted mice, SCFA receptor deficiency rendered SPF mice more susceptible to *K. pneumoniae*, and microbiota depletion also impaired bacterial control during the infection (Fig. 3A). Oral treatment with SCFAs, i.e. acetate (Ace), propionate (Pro) or butyrate (But) partially rescued antibacterial defense in microbiota-depleted WT animals through FFAR2/FFAR3 (Fig. 3B, C). Assuming that the differential bacterial loads at the 24 h time point might be explained by attenuated immune responses before, we next examined inflammatory responses 12 h after infection, when bacterial loads were not yet different (Fig. S3A). With the exception of a small difference in IL-10, we did not observe any significant impact of FFAR2/FFAR3 deficiency on cytokines and chemokines levels (Fig. 3D, Fig. S3B), nor did we observe an effect of microbiota depletion (Fig. S3C). In line with these results, numbers and

**Table 1 | Summary of the patient cohorts**

| | Male patients w/o ABX ($n = 23$) | Female patients w/o ABX ($n = 13$) | Male patients with ABX ($n = 25$) | Female patients with ABX ($n = 11$) | |
|---|---|---|---|---|---|
| Age [years] | 68.56 ± 9.12 | 58.08 ± 12.89 | 66.6 ± 9.87 | 59.75 ± 12.61 | $p > 0.05$ |
| Female [$n$, %] | 13 (36.1) | | 11 (30.5) | | $p > 0.05$ |
| BMI | 26.05 ± 5.3 | 25.36 ± 6.72 | 24.13 ± 4.25 | 22.40 ± 5.11 | $p > 0.05$ |
| CRP [mg/dL] | 4 (1.80–11.80)* | 7.5 (2.5-13.75) | 26.5 (8.0-100.7) | 12.3 (4.0-50.8) | *$p < 0.005$ |
| WBC [G/L] | 8.23 (6.23-10.02) | 8.14 (6.60-9.47) | 9.59 (7.35-11.72) | 8.13 (6.78–12.1) | $p > 0.05$ |
| Main current antibiotics | | | | | |
| - carbapenem | 0 (0) | 0 (0) | 15 (60) | 3 (27.27) | |
| - tazobactam | 0 (0) | 0 (0) | 11 (44) | 7 (63.63) | |
| - ampicillin/sulbactam | 0 (0) | 0 (0) | 1 (4) | 1 (9.09) | |
| - ceftriaxone | 0 (0) | 0 (0) | 1 (4) | 0 () | |
| - fluoroquinolone | 0 (0) | 0 (0) | 1 (4) | 0 () | |
| - TMP/SMX | 0 (0) | 0 (0) | 1 (4) | 2 (18.18) | |
| - macrolides | 0 (0) | 0 (0) | 1 (4) | 1 (9.09) | |
| Form of nutrition | | | | | |
| - per os only | 23 (100) | 13 (100) | 23 (92) | 9 (81.81) | |
| - PEG/gastric tube | 0 (0) | 0 (0) | 2 (8) | 2 (18.19) | $p > 0.05$ |
| Reasons for hospital admission | | | | | |
| - chronic lung disease | 16 (69.56) | 9 (69.23) | 9 (36) | 3 (27.27) | $p > 0.05$ |
| - acute deterioration of chronic lung disease | 2 (8.69) | 0 (0) | 6 (24) | 2 (18.18) | $p > 0.05$ |
| - acute lung infection | 0 (0)* | 1 (7.69) | 10 (40)* | 4 (36.36) | *$p < 0.001$ |
| - ventilatory failure due to neurological disease | 2 (8.69) | 1 (7.69) | 0 (0) | 0 (0) | $p > 0.05$ |
| - other | 3 (13.6) | 2 (15.38) | 0 (0) | 2 (18.18) | $p > 0.05$ |
| Comorbidities | | | | | |
| - diabetes mellitus | 7 (30.43) | 0 (0) | 9 (36) | 1 (9.09) | |
| - chronic liver disease | 4 (17.39) | 0 (0) | 5 (20) | 0 (0) | |
| - IBD/acute colitis | 0 (0) | 0 (0) | 0 (0) | 0 (0) | $p > 0.05$ |

*BMI* body mass index, *CRP* C-reactive protein, *IBD* inflammatory bowel disease, *PEG* percutaneous endoscopic gastrostomy tube, *TMP/SXT* co-trimoxazole, *WBC* white blood cell count. Kruskal-Wallis test followed by Dunn´s multiple comparison was applied to the dataset.

percentage of major resident and recruited immune cells including alveolar macrophages (AMs), polymorphonuclear neutrophils (PMNs) and IMs did not differ between WT and *Ffar2/Ffar3*[-/-] mice (Fig. 3E, F, Fig. S4A–C). Collectively, inflammatory cytokine production and recruitment of immune cells classically associated with antibacterial defense were not affected by FFAR2/FFAR3 deficiency.

### FFAR2/FFAR3 deficiency influences gene expression in PMNs and IMs

Considering that FFAR2/3 deficiency affects *K. pneumoniae* infection but apparently not the recruitment of major antibacterial immune cells, we hypothesized that it affects the activation status and gene expression of specific immune cells instead. To test this hypothesis, single-cell RNA sequencing (scRNAseq) was performed to determine how gene expression in pulmonary cells is influenced by lack of SCFA receptors. Lungs of four mice per group of *K. pneumoniae* ST258-infected or PBS-treated WT and *Ffar2/Ffar3*[-/-] animals were collected and single-cell suspensions of pooled cells were subjected to scRNA library preparation. A total of 30,215 cells were analyzed, and after UMAP dimensionality reduction, 18 distinct cell types were identified that were characterized by typical marker gene expression (Fig. S5A-C). We next focused on AMs, PMNs and monocytes as important pulmonary immune cells in the early phase of pneumonia. For each of these cell types, we defined two individual cell subsets (Fig. 4A, B, D, E, G, H) which in case of monocytes represent IMs and patrolling monocytes (PMs) (Fig. 4G, H), as indicated by expression of characteristic marker genes (Fig. S5D). Differential gene expression

analyses of individual cell subsets revealed that the SCFA receptors strongly affect PMNs and IMs (Fig. 4C, F, I). For example, genes encoding proteins involved in lysosomal degradation such as *Lyz2* and *Psap* (Fig. 4I) as well as corresponding gene sets (Fig. 4J) were downregulated in IMs from *Ffar2/Ffar3*[-/-] mice. In contrast, little or no significantly differentially expressed genes were identified in AMs and PMs, possibly partly due to the lower cell numbers analyzed. Overall, our analyses suggest that SCFA affect gene expression and possibly antibacterial activity of IMs during bacterial infection.

### Role of IMs in FFA2/FFAR3-dependent pulmonary antibacterial defense

Next, we determined if AMs, PMNs or IMs were involved in the SCFA-controlled antibacterial defense against *K. pneumoniae* lung infection. First, we demonstrate that treatment with SCFA was able to improve pulmonary antibacterial defense also in *Csf2*[-/-] mice that lack AMs but not PMNs and IMs (Fig. 5A, Fig. S6A–C)[24,25]. Second, we examined possible involvement of PMNs by depleting them in both WT and *Ffar2/Ffar3*[-/-] mice. We observed that SCFA receptor deficiency compromised antibacterial control in mice after antibody-mediated reduction of PMNs to approximately the same extent as in mice treated with control antibodies (Fig. 5B, Fig. S6D–F), suggesting that the SCFA-dependent protective effect in *K. pneumoniae* lung infection is primarily mediated by other cell types. Third, we investigated potential involvement of IMs by depleting them with anti-CCR2 antibodies in WT and *Ffar2/Ffar3*[-/-] mice. Consistent with previous studies reporting the ability of IMs to kill *K. pneumoniae* during

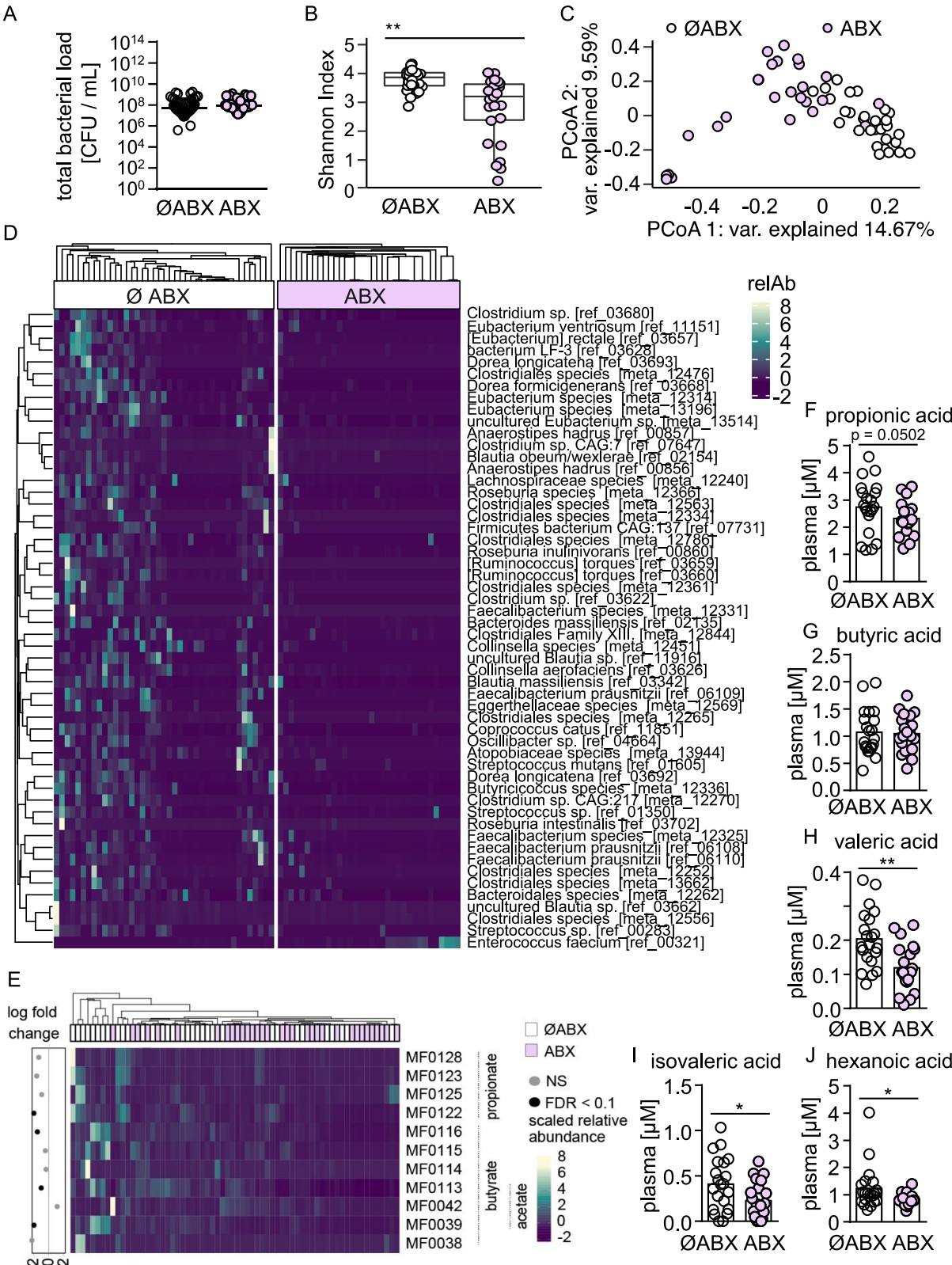

lung infections[26,27], depletion of IMs impaired pulmonary control of *K. pneumoniae* infection in WT mice. Interestingly, however, anti-CCR2 treatment did not further attenuate antibacterial defenses in mice lacking SCFA receptors (Fig. 5C, Fig. S6G–I). Moreover, we compared the capacity of adoptively transferred bone-marrow isolated IMs from WT and *Ffar2/Ffar3⁻/⁻* mice to resuce antibacterial

resistance in *Ccr2⁻/⁻* animals, which lack the ability to recruit their own IMs into the lungs. We observed that *Ccr2⁻/⁻* mice receiving WT IMs had lower bacterial loads as compared to *Ccr2⁻/⁻* mice receiving *Ffar2/Ffar3⁻/⁻* IMs or PBS (Fig. 5D, Fig. S6J). Collectively, our results demonstrate that IMs critically contribute to the protective effect of SCFA sensing in *K. pneumoniae* infections.

**Fig. 1 | Antimicrobial therapy in patients is associated with reduced diversity of the gut microbiota and decreased abundance of SCFA producers. A** Total bacterial loads (CFU) in fecal samples of antibiotic-treated (ABX; $n = 12$) and untreated patients (ØABX; $n = 17$). **B–E** Shotgun metagenomic sequencing was conducted on fecal samples of antibiotic-treated (ABX; $n = 26$) and untreated (ØABX; $n = 29$) patients. **B** α-diversity (Shannon index) and (**C**) PCoA of beta diversity were calculated on pairwise Bray-Curtis dissimilarities. **D** Heatmap indicating the scaled relative abundance of bacterial species altered in antibiotic-treated patients compared to non-antibiotic treated controls. **E** Abundances of gene modules associated with the production of SCFAs in fecal samples of antibiotic-treated and untreated patients. **F–J** SCFA concentrations in plasma samples of antibiotic-treated (ABX; $n = 21$) and untreated (ØABX; $n = 21$) patients. Mann-Whitney $U$ test (two-tailed) was used for bacterial loads (**A, B**) and Wilcoxon-Mann-Whitney $U$ test (two-tailed) was used for plasma SCFA concentrations (**F–J**) and values are shown as median with each dot representing the data from one patient. **B** The box plot represents median, 25th and 75th percentiles—interquartile range; IQR—and whiskers extend to maximum and minimum values *$P < 0.05$, **$P < 0.01$, ***$P < 0.005$.

## Antibiotic-induced microbiota alterations compromise FFAR2/FFAR3-controlled antibacterial activity of IMs

Microbicidal activity of IMs are required to clear pulmonary *K. pneumoniae* infections[27]. To determine if SCFA directly affect IM´s microbicidal activity, IMs were isolated from bone marrow of WT and *Ffar2/Ffar3−/−* mice, treated with SCFAs or left untreated, and co-incubated with *K. pneumoniae* for 2 hrs. We found that SCFA enhanced antibacterial activity of WT but not *Ffar2/Ffar3−/−* IMs (Fig. 6A). Moreover. the SCFA-induced enhancement of IM´s microbicidal activity was blocked the phagocytosis inhibitor cytochalasin D, the endolysomal acidification blocker bafilomycin A1, and the lysozyme inhibitor N,N',N''-Triacetylchitotriose (Fig. 6A). To test if antibiotic-induced gut microbiota alterations affect IM´s antibacterial activity, the cells were isolated from bone marrow of mice transplanted with either individual antibiotic-naïve or antibiotic-associated patient microbiota. IMs from WT mice transplanted with antibiotic-naïve microbiota showed antibacterial activity against *K. pneumoniae*, whereas IMs from WT mice transplanted with antibiotic-associated microbiota or from *Ffar2/Ffar3−/−* mice were unable to kill *K. pneumoniae* (Fig. 6B). Together, these results demonstrate that antibiotic-induced microbiota alterations compromise SCFA-controlled microbicidal activity of IMs. SCFA enhance antibacterial activity of IMs by a mechanism involving FFAR2/FFAR3, endolysosoaml acidification and lysozyme activity.

## Discussion

Here, we characterize how clinical antibiotic therapy influences pulmonary antimicrobial immune defense in the context of human gut microbiota. We reveal that antibiotic-induced depletion of SCFA producers affects microbicidal activity of IMs, thereby compromising antibacterial immune defense in the lung. Our results thus propose explanation for the well-known association between antimicrobial therapy and subsequent HAP[4–7,9], provide additional arguments for the prudent use of antibiotics, and offer mechanistic insights for the development of novel prophylactic strategies to protect high-risk patients from HAP.

Antibiotic therapy is the cornerstone of treating bacterial infections, and recent point-prevalence studies indicated that the majority of ICU patients receive antibiotics[28,29]. Unfortunately, antibiotics target not only the invasive pathogen but, inevitably, also the microbiota, which in turn impairs colonization resistance and favors intestinal colonization with opportunistic pathogens such as *K. pneumoniae*[11–13]. We show here that therapy with widely used parentally given broad-spectrum antibiotics (almost all of our included patients received either carbapenems or piperacillin-tazobactam) additionally compromises pulmonary antimicrobial immune response by depleting SCFA producers and inhibiting IMs. While our knowledge of the effects of different classes of antibiotics on the microbiome is still relatively sparse, our findings are consistent with recent reports indicating that particularly both piperacillin-tazobactam and carbapenems severely disrupt the gut microbiota and decimate anaerobic commensals[9,30,31]. How other classes of antibiotics affect microbiome-dependently controlled immune mechanisms is currently unknown. Future studies are therefore needed to characterize and compare the collateral damage to the microbiota and immune system by different antibiotics. It is, however, reasonable to speculate that antibiotics with a narrower spectrum and less activity against anaerobic bacteria may have less dramatic effects on the microbiota-controlled defense mechanisms. If this assumption is correct, then another important argument would emerge for the use of narrow-spectrum antibiotics in the targeted therapy of bacterial infections whenever possible.

Our study may help to develop new prophylactic strategies to protect high-risk patients, such as those receiving broad-spectrum antibiotics, from HAP. While a number of previous small and heterogeneous randomized controlled trials suggested that probiotics reduce HAP incidence[32], the recent large multicenter PROSPECT trial involving 2,650 patients did not find evidence of a protective effect of probiotics[33]. However, in most of these studies including the PROSPECT trial, *Lactobacillus rhamnosus* GG was administered as probiotic, and it remained unclear to what extend *L. rhamnosus* GG successfully colonized the intestine and whether it was able to strengthen antimicrobial defense mechanisms. Future research is needed to develop next-generation probiotics or perhaps other strategies to enhance the microbicidal activity of IMs and other critical immune cells in patients receiving broad-spectrum antibiotics.

A large variety of studies provide important insights into the interaction between the microbiota and the immune system using germ-free mice or animals treated with a cocktail of oral antibiotics for several weeks[18,20–22]. Our study adds to this by characterizing how parenteral antibiotic therapy typically used in hospitalized patients affects the human microbiota and how this influences antimicrobial immunity during infection.

SCFAs are microbiota-derived metabolites that influence various physiological processes such as antimicrobial immune defense through FFARs or histone deacetylase (HDAC) inhibition[34–37]. Previous work, for example, showed that SCFAs modulate reactive oxygen species production, phagocytosis and chemotaxis of PMNs through FFAR2[38], enhance generation of PMs via FFAR3[39], and imprint an antimicrobial program in macrophages via their HDAC3 inhibitory function[35,40]. We add to the knowledge by demonstrating that microbicidal activity of IMs is enhanced by SCFAs via FFAR2/3, phagocytosis, endolysosomal acidification, and lysozyme activity, and that this mechanism critically contributes to the control of bacterial lung infections in vivo. The reduced microbicidal activity of IMs of mice transplanted with antibiotic-associated microbiota or animals lacking FFAR2/FFFAR3 correlated with decreased expression of genes linked to phagocytosis, lysosomal degradation, and lysozyme activity. Future studies are, however, required to further explore the molecular pathways by which antibiotics-induced microbiota disturbance affects IMs´ antibacterial activity.

Our study has several limitations. First, our study lacks statistical power to discriminate between the effects of different classes of antibiotics, as most of our patients received either carbapenems or piperacillin-tazobactam. However, these beta-lactams represent highly relevant antibiotics that are commonly used in hospitalized patients. Second, we did not directly examine antibacterial activity of IMs from patients receiving antibiotics. To exclude confounders including the

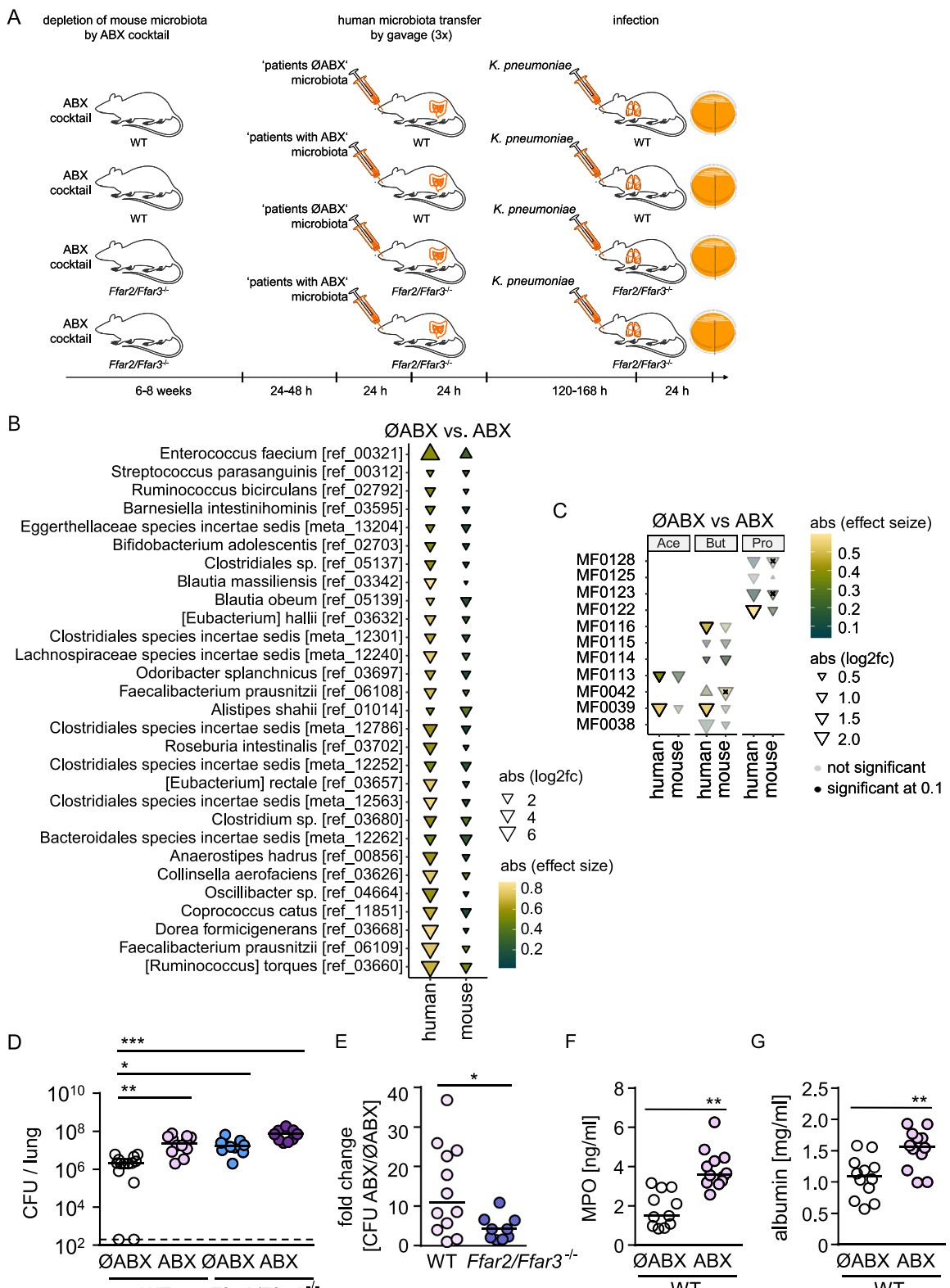

primary infection requiring the antibiotic therapy and comorbidities, such study should be conducted in the future in healthy subjects receiving antibiotics or placebo. Third, we cannot rule out, but rather consider it likely, that broad-spectrum antibiotic therapy also compromises additional microbiota-controlled immune mechanisms, whose impairment could contribute to the patient's HAP predisposition. However, the work presented here demonstrates how clinically relevant parenterally given antibiotics affect IM-dependent antibacterial defense in the context of human microbiota. It offers a mechanism underlying the association between antimicrobial therapy and subsequent HAP[4–7,9], might guide antibiotic stewardship measures in the clinic to reduce antibiotics´ collateral damage on microbiota-controlled defense mechanisms, and could help to develop novel prophylactic strategies against HAP.

**Fig. 2 | Human microbiota alterations induced by antibiotics or FFAR2/FFAR3 deficiency compromise pulmonary antibacterial defense in mice.**
**A** Experimental scheme of human microbiota transplantation and infection experiments for **B**–**E**. Antibiotic-associated alterations in the relative abundance of bacterial species and functional modules in fecal samples of transplanted mice and patient samples, significantly altered species with similar directionality are shown (**B**), and variation in SCFA modules (**C**) (n = 26 for ØABX, n = 29 for ABX). **D** Bacterial loads (CFU) in lungs of human microbiota transplanted mice infected with *K. pneumoniae* for 24 h, and **E** fold changes in lung bacterial loads of mice transplanted with ABX-associated human microbiota vs. ABX-naïve human microbiota (n = 12 for ØABX WT; n = 12 for ABX WT; n = 11 for ØABX *Ffar2/Ffar3*[-/-]; n = 9 for ABX *Ffar2/Ffar3*[-/-]). **F, G** Myeloperoxidase and albumin levels in bronchoalveolar lavage fluid (BALF) of infected mice, transplanted with ABX-naïve human microbiota or ABX-associated human microbiota (n = 12 for ØABX WT; n = 12 for ABX WT). **C** FDR correction was done to account for multiple testing using the Benjamini-Hochberg method (two-sided). Kruskal-Wallis test followed by Dunn´s multiple comparison was applied to the bacterial load dataset (**D**). The fold change analysis (**E**) and myeloperoxidase (MPO) and albumin measurements (**F**–**G**) were analyzed by Mann-Whitney *U* test (two-tailed). Values are shown as median (**D**–**G**), each dot represents the data from a single mouse. The solid black border of triangles in C indicates q < 0.1 (FDR corrected in the human samples, two-sided), and x on the triangles indicates p < 0.05 (targeted testing only within the significant modules in human samples). *P < 0.05, **P < 0.01, ***P < 0.005. Downward facing triangle: increased in ØABX in comparison to ABX.

## Methods

### Study approval
All animal experiments were carried out in adherence to the German Animal Welfare Act (Tierschutzgesetz, TierSchG) and to the Federation of European Laboratory Animal Science Association (FELASA) guidelines, following approval by the responsible institutional (Charité – Universitätsmedizin Berlin) and governmental animal welfare authorities (LAGeSo Berlin, approval ID G0284/16 and G0010/22). The human observational study was approved by the institutional review board (Ethics Committee of the Charité – Universitätsmedizin Berlin, identifier: EA4/232/19) and was registered at the clinical study registry of the Charité – Universitätsmedizin Berlin (https://studienregister.charite.de, identifier: 3000023). Written informed consent was received from all participants, and no compensation was paid. Sex of participants are indicated but not further considered in the study design as we describe a general mechanism that is likely not to be sex/gender-specific.

### Patient stool and plasma sampling
The cohort of patients was recruited between 2020 and 2021 in respiratory medicine wards at Charité. Patients gave their informed consent for sampling. Inclusion criteria were age ≥ 18 years, at least two days of hospitalization and therapy with broad-spectrum antibiotics (with activity against Gram-positive, Gram-negative, and anaerobic bacteria), or no antibiotic use for at least 2 months prior to participation in the study. Patients meeting the following criteria were excluded from the study: inability to provide consent to participate in the study, involuntary hospitalization under the mental health act (PsychKG), acute *Clostridioides difficile* infection, chronic inflammatory bowel disease and cystic fibrosis. Stool samples were collected from patients and stored at −4 °C for a maximum of 24 h, before being transferred to storage at −80 °C. Blood samples were collected in tubes containing EDTA and spun down at 500 g for 10 min to obtain plasma, which was stored at −80 °C.

### Mice
Male and female *Ffar2/Ffar3*[-/-41], *Csf2*[-/-] (Jackson Laboratory), and *Ccr2*[-/-] (Jackson Laboratory) mice on C57BL/6 background used for comparative analysis and experiments were housed & bred in the same facilities (animal husbandry, Charité) with identical specific pathogen-free conditions under 22 °C, 50 − 55% relative humidity, and 12 h/12 h light/dark cycle conditions as C57Bl/6 WT control animals, with free access to food (rat/mouse maintenance food V1534-000, Ssniff) and water. Sex was not considered in this study. Mice were chosen by fitting age ranging from 8 to 16 weeks.

### Gut microbiota depletion and SCFA treatment
For the depletion of the intestinal mouse gut microbiota, 8-9-week-old mice were housed in sterile cages and treated orally by adding imipenem (250 mg/L; Fresenius Kabi), metronidazole (1 g/L; Braun), vancomycin (500 mg/L; HIKMA Pharma), ciprofloxacin (200 mg/L;

Fresenius Kabi) and ampicillin (1 g/L; Ratiopharm) to the autoclaved drinking water *ad libitum* for six weeks (ABX mice). The antibiotic containing drinking water was renewed every three days of treatment. During treatment, fecal samples were collected weekly for monitoring of potential fungal outgrowth and microbiota depletion progress by bacterial and fungal cultivation on agar plates. Mice with fungal outgrowth were excluded from the experiments. Animals used as conventional microbiota controls (CONV) for ABX mice were housed in the same animal facilities and rooms under specific pathogen-free conditions synchronously as ABX animals were undergoing antibiotic treatment. In some experiments, acetate, sodium butyrate or sodium propionate or sodium chloride (Sigma-Aldrich) as control were additionally administered into the drinking water in concentrations of 200 mM during the last two weeks of ABX treatment until day of preparation of mice. The SCFAs or sodium chloride in the drinking water were replenished every three days. For the infection of SCFA- or sodium chloride-treated ABX mice, the drinking water containing the antibiotics mixture was changed to autoclaved drinking water substituted with one of the sterile filtered fatty acids or sodium-chloride 2 to 3 days prior intranasal infection.

### K. pneumoniae infection
Kp235-11 is a KPC-2 carbapenemase-producing *K. pneumoniae* clinical isolate (sequence type ST258), which was kindly provided by Dr. Yvonne Pfeifer, Robert Koch-Institute, Wernigerode, Germany, and has been deposited at Leibniz Institute DSMZ, Germany (DSM 113479). Kp235-11 was cultured overnight in tryptic soy broth (TSB) at 37 °C with an agitation speed of 200 rpm under aerobic conditions. After overnight culture, the bacteria were washed with sterile PBS and adjusted for the infection dose ($1 \times 10^7$ CFU/mouse in the PMN depletion experiments, $1 \times 10^8$ CFU/mouse in all experiments) on the optical density at $OD_{600nm}$ and administered intranasally into anesthetized mice with 20 µL in one nostril. Control groups received sterile phosphate-buffered saline. For infection of ABX animals, the drinking water containing the antibiotics was changed to autoclaved drinking water 2-3 days prior infection. Infection doses were controlled by serial dilution and plating on blood agar in duplicates. At the indicated time points of 12 h and 24 h post infection, mice were euthanized by an intraperitoneal injection of ketamine and xylazine and by drawing blood from the *vena cava caudalis*. After euthanizing, BALF with cold PBS was performed; whole lungs were harvested afterwards, homogenized in sterile PBS and filtered through a 70 µM cell strainer[42]. Bacterial burdens were quantified by plating serial dilutions of lung homogenates on blood agar plates and colonies were counted after plates were incubated overnight with 5% $CO_2$ at 37 °C. BALF and remaining homogenized lungs were centrifuged with a speed of 1500 g at 4 °C for 10 minutes to part cells from supernatants. The separated BAL supernatants were frozen in liquid nitrogen for cyto- and chemokine measurements and separated lung homogenates were used for further FACS analysis.

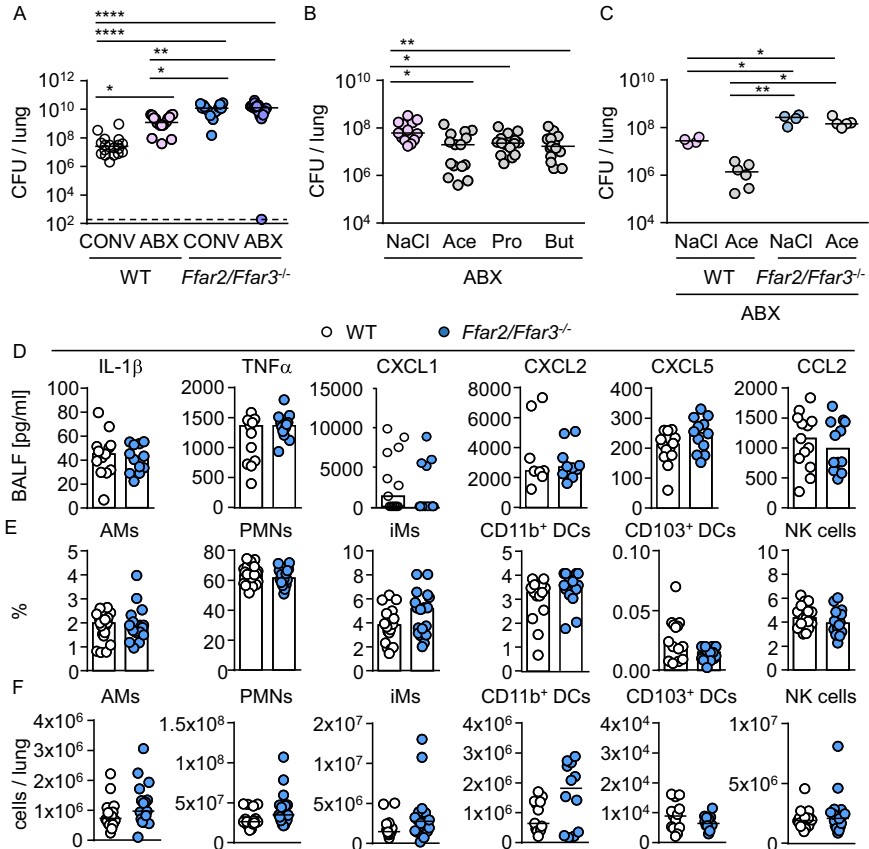

**Fig. 3 | SCFA receptor deficiency does not influence pulmonary cytokine production and immune cell infiltration during *K. pneumoniae* infection.**
**A** Conventionally colonized (CONV) or antibiotic (ABX)-treated WT and *Ffar2/Ffar3⁻/⁻* mice were infected with *K. pneumoniae* for 24 h and bacterial loads (CFU) in lungs were assessed (*n* = 18 for CONV WT; *n* = 18 for ABX WT; *n* = 17 for CONV *Ffar2/Ffar3⁻/⁻*; *n* = 17 for ABX *Ffar2/Ffar3⁻/⁻*). **B** WT mice treated with antibiotics (ABX) were additionally given SCFA or NaCl. Lung bacterial loads were assessed after 24 h (*n* = 13 for NaCl; *n* = 15 for Ace; *n* = 15 for Pro; *n* = 15 for But). **C** WT and *Ffar2/Ffar3⁻/⁻* mice treated with antibiotics (ABX) were additionally given acetate (200 mM, Ace) or NaCl, and lung bacterial loads were assessed (*n* = 4 for NaCl WT; n = 6 for Ace WT;

*n* = 4 for NaCl *Ffar2/Ffar3⁻/⁻*; *n* = 5 for Ace *Ffar2/Ffar3⁻/⁻*). **D–F** Conventionally housed mice were intranasally infected with *K. pneumoniae* for 12 h, cytokine levels in BALF were measured by multiplex ELISA (*n* = 12 for WT; *n* = 13 for *Ffar2/Ffar3⁻/⁻*) or AMs (alveolar macrophages), PMNs (pulmonary neutrophils), IMs (inflammatory monocytes), CD11b⁺ DCs (dendritic cells), CD103b⁺ DCs, and NK cells (natural killer cells) in lung tissue were analyzed by FACS (*n* = 19 for WT; *n* = 18 for *Ffar2/Ffar3⁻/⁻*). Kruskal-Wallis test followed by Dunn´s multiple comparison was applied to the bacterial load datasets (**A–C**). Mann-Whitney *U* test (two-tailed) was applied for lung cell populations and cytokines (**D–F**). Values are shown as median, each dot represents the data from one mouse. *$P < 0.05$, **$P < 0.01$, ***$P < 0.005$.

## Cytokine, chemokine, albumin, and MPO measurements
Cytokine and chemokine levels in collected BAL samples were measured by using custom-designed multiplex antibody- and magnetic bead-based protein quantification assays (ProcartaPlex; purchased from Thermo-Fisher) according to manufacturer´s instructions. Myeloperoxidase (MPO) and albumin levels in collected BALF samples were measured by using Mouse Myeloperoxidas Elisa Kit (Abcam) and Mouse Albumin Elisa Kit (Bethyl Laboratories) according to manufacturer´s instructions.

## Flow cytometry
Homogenized lungs were incubated in DNAse/Collagenase at 37 °C for 40 minutes, followed by filtering of cell suspensions through a 70 µM cell strainer and red blood cell lysis to obtain a single cell suspension. Cells were stained with LIVE/DEAD Fixable Cell Dead stain (diluted 1/1000, Thermo Fisher Scientific) and with different combinations of the following anti-mouse antibodies: CD11b-BV421 (diluted 1/250, clone M1/70, BioLegend), Ly6G-BV510 (diluted 1/300, clone 1A8, BioLegend), CD45-FITC (diluted 1/200, clone 30-F11, BioLegend), Ly6C-PerCP-Cy5.5 (diluted 1/200, clone HK1.4, BioLegend), Siglec-F-PE (diluted 1/150, clone E50-2440, Becton Dickinson), CD11c-APC (diluted 1/300, clone APC-N418, BioLegend), CD4-BV510 (diluted 1/200, clone RM4-5,

BioLegend), γδTCR-FITC (diluted 1/150, clone GL3, Becton Dickinson), CD45-PerCP-Cy5.5 (diluted 1/200, clone 30-F11, Becton Dickinson), CD3-PE (diluted 1/200, clone 17A2, Becton Dickinson), NK1.1-PECy7 (diluted 1/200, clone PK136, BioLegend), CD24-BV450 (diluted 1/200, clone M1/69, Becton Dickinson), CD11b-BV510 (diluted 1/100, clone M1/70, BioLegend), CD11c-FITC (diluted 1/150, clone N418, BioLegend), CD64-PE (diluted 1/200, clone X54-5/7.1, BioLegend), MHCII-PECy7 (diluted 1/200, clone M5/114.15.2, BioLegend) and CD103-APC (diluted 1/200, clone M290, Becton Dickinson). For cell counting, CountBright™ absolute counting beads (Molecular Probes) were added to the samples prior to acquisition in a FACSCanto II analyzer (Becton Dickinson). The acquired data was analyzed with FlowJo V10 software (TreeStar). Gating strategies were processed after excluding cell doublets and dead cells.

## Cell depletion
To deplete PMNs in vivo, anti-Ly6G (clone 1A8; #BP0075-1), anti-rat kappa immunoglobulin light chain (clone MAR 18.5; #BE0122), and isotype control rat IgG2a (clone 2A3; # BP0089) purchased from Bio X Cell were injected intraperitoneally 2 h prior infection of the animals in doses according to the mice's body weight of 1 µg/g of anti-Ly6G and 2 µg/g of anti-rat Kappa immunoglobulin light chain or 1 µg/g of the isotype control. For the depletion of monocytes, mice were injected

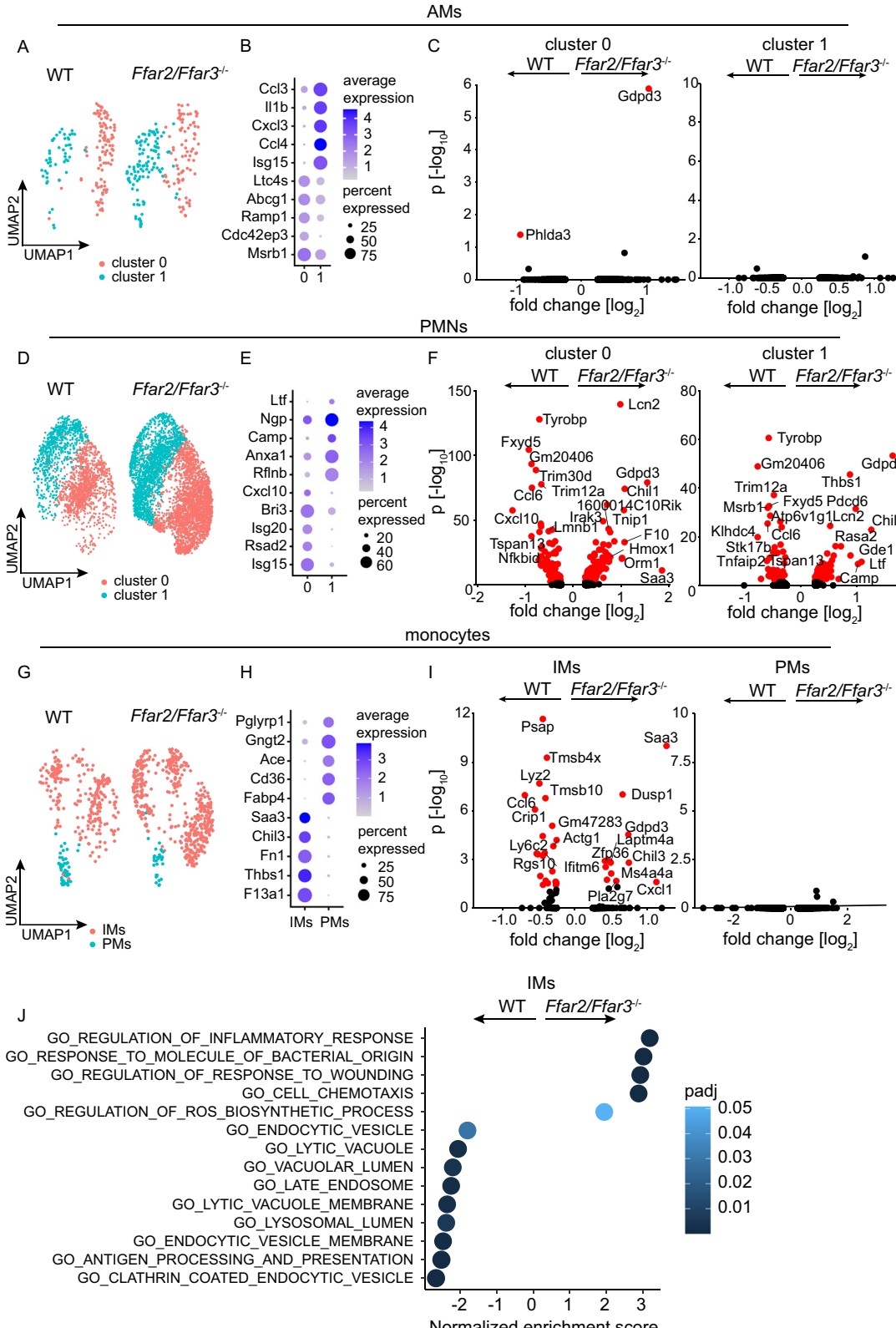

**Fig. 4 | FFAR2/FFAR3 deficiency affects gene expression in IMs (inflammatory monocytes) and PMNs (pulmonary neutrophils).** scRNAseq of lung cells from WT and *Ffar2/Ffar3*⁻/⁻ mice infected with *K. pneumoniae* for 12 h (*n* = 4 for WT, *n* = 4 for *Ffar2/Ffar3*⁻/⁻). **A**, **D**, **G** UMAPs of AMs, PMNs, and monocytes subclusters. **B**, **E**, **H** Dot plots of the most differentially expressed genes for each cell subcluster. **C**, **F**, **I** Volcano plots of differentially expressed genes for each cell subclusters.

**J** GSEA indicating significantly up- and downregulated pathways in IMs of infected WT and *Ffar2/Ffar3*⁻/⁻ mice. Wilcoxon Rank Sum test was used and adjusted based on bonferroni correction using all genes in the dataset. **J** Bar plot indicating significantly up- and downregulated GSEA pathways in IMs of infected WT and *Ffar2/Ffar3*⁻/⁻ mice. Calculation of p-value estimation is based on an adaptive multi-level split Monte-Carlo scheme followed by BH correction.

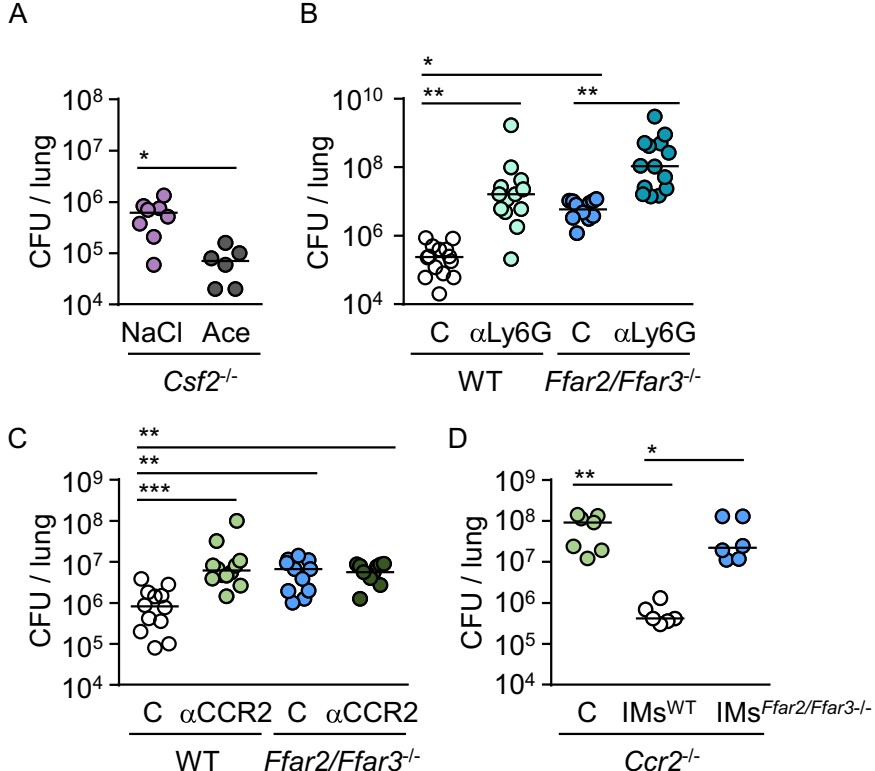

**Fig. 5 | FFAR2/FFAR3 control antibacterial activity via IMs. A** Bacterial loads (CFU) in lung tissue of ABX-treated *Csf2*[-/-] mice receiving either NaCl or acetate (*n* = 8 for NaCl, *n* = 6 for Ace) 24 h after infection with *K. pneumoniae*. **B** Bacterial loads in WT and *Ffar2/Ffar3*[-/-] mice treated intraperitoneally with anti-Ly6G or control antibody (**C**) and infected for 24 h with *K. pneumoniae* (n = 14 for WT C, *n* = 13 for *Ffar2/Ffar3*[-/-] C, *n* = 13 for WT αLy6G; *n* = 14 for *Ffar2/Ffar3*[-/-] αLy6G). **C** Bacterial loads in WT and *Ffar2/Ffar3*[-/-] mice treated intraperitoneally with anti-CCR2 or control antibody (**C**) and infected for 24 h with *K. pneumoniae* (n = 12 for

WT C, *n* = 11 for *Ffar2/Ffar3*[-/-] C, *n* = 12 for WT αCCR2; *n* = 11 for *Ffar2/Ffar3*[-/-] αCCR2). **D**) Bacterial loads in the lungs of *Ccr2*[-/-] mice transplanted intravenously with IM´s of WT or *Ffar2/Ffar3*[-/-] animals or treated with PBS, and infected with *K. pneumoniae* (*n* = 7 for PBS, *n* = 6 for WT; *n* = 6 for *Ffar2/Ffar3*[-/-]). Kruskal-Wallis test followed by Dunn´s multiple comparison was applied to the datasets. Values are shown as median, each dot represents the data from a single mouse. *\*P* < 0.05, *\*\*P* < 0.01, *\*\*\*P* < 0.005.

with 20 µg/mouse of rat anti-mouse CCR2 (clone MC-21, gifted by Matthias Mack) or corresponding isotype control IgG2b (clone RTK4530; #400666, BioLegend) intraperitoneally immediately at the time of infection.

### Bone marrow isolation
Mice were sacrificed, hind limbs were separated and cleaned from surrounding tissues and dipped in ice-cold ethanol. Femur and tibia were disconnected, and bone marrow was collected by cutting off the ends and by flushing the bones with 10 mL sterile PBS. Isolated bone marrow was filtered with a 70 µM cell strainer, spun down at 500 g for 10 min at 4 °C, and red blood lysis was performed.

### FACS sorting
Bone marrow suspensions were stained with CD11b-PECy7 (diluted 1/100, clone M1/70, BioLegend), Ly6G-PE (diluted 1/100, clone 1A8, BioLegend), CD45-FITC (diluted 1/100, clone 30-F11, BioLegend), Ly6C-PerCP (diluted 1/100, clone HK1.4, BioLegend), and NK1.1-APC (diluted 1/100, clone S17016D, BioLegend) followed by washing and sorting of the cells using a FACS Aria™ II SORP flow cytometer cell sorter (Becton Dickinson). Sorted IMs were defined as Ly6G-NK1.1-CD11b+Ly6C(hi) cells and used for killing assays or were frozen in TRIzol at −80° for RNA isolations.

### Bacterial killing assays
Kp235-11 was opsonized by incubation with 30 µl mouse serum for 30 minutes at 37 °C. FACS-sorted IMs were counted, spun down at

500 g for 10 min at 4 °C, resuspended in RPMI containing 2% fetal calf serum, and seeded in triplicates on 96 well plates. In some experiments, IMs were incubated for 20 min with 1 mM acetate (Sigma-Aldrich) or 1 mM acetate in combination with 50 µM bafilomycin A1 (Biomol), 0.75 µg cytochalasin D (Sigma-Aldrich) or 100 µg N,N´,N″-Triacetylchitotriose (Sigma-Aldrich) and washed twice prior to incubation with bacteria. Cell suspensions were then incubated with opsonized Kp235-11 in a MOI of 0.01 at 37 °C. After 30 or 120 min, 10-fold dilutions of the cellular supernatants were plated on blood agar plates, and CFUs were counted after overnight incubation.

### Human fecal microbiota transplantation
Microbiota-depleted mice were transplanted with fecal material from our patient cohort. Briefly, thawed stool samples from each individual donor were resuspended in sterile PBS, homogenized and filtered using a 70 µM cell strainer. 300 µl of the resuspended fecal material was administered to WT and *Ffar2/Ffar3*[-/-] mice three times on three consecutive days by oral gavage. Stool samples were transplanted individually (not pooled), so that in most cases each pair of WT and one *Ffar2/Ffar3*[-/-] mouse received the sample of one individual donor. Five to eight days after fecal transplantation, mice were infected with *K. pneumoniae* or used for isolation of IMs. Fecal samples from transplanted animals were collected for subsequent metagenomics sequencing.

### Adoptive cell transfer
For the adoptive cell transfer of IMs, donor WT and *Ffar2/Ffar3*[-/-] mice were sacrificed and bone marrows were isolated. Cells were stained

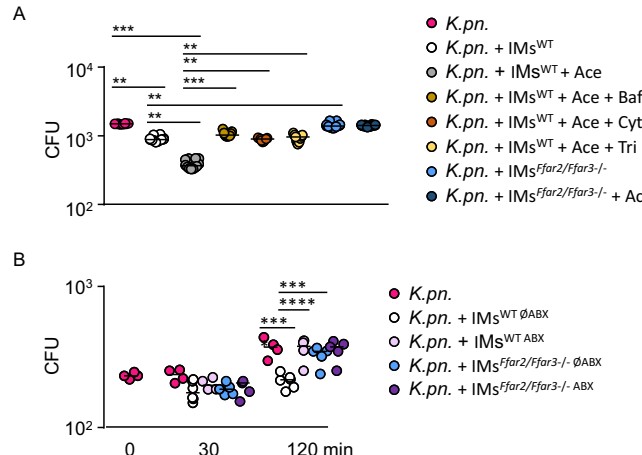

**Fig. 6 | Antibiotic-induced microbiota alterations compromise FFAR2/FFAR3-controlled antibacterial activity of IMs (inflammatory monocytes). A** IMs of WT and *Ffar2/Ffar3*[-/-] mice were isolated from the bone marrow by FACS sorting, stimulated for 20 minutes with 1 mM acetate (Ace) or control medium prior to incubation with *K. pneumoniae*. WT IMs were also stimulated with 1 mM acetate in combination with 50 μM bafilomycin A1 (Baf), 0.75 μg cytochalasin D (Cyto), or 100 μg N,N´,N″-Triacetylchitotriose (Tri) prior to incubation with the bacteria. Values are shown as median, each dot represents the mean data from triplicates from cells of one mouse (*n* = 8 for *K. pneumoniae* without IMs; n = 12 for IM[WT]; *n* = 12 for IM[WT] + Ace; n = 9 for IM[WT] + Ace + Baf; *n* = 9 IM[WT] + Ace + Cyto; n = 9 for IM[WT] + Ace + Tri; n = 12 for IMFfar2/Ffar3-/-; *n* = 12 for IM[*Ffar2/Ffar3-/-*] + Ace). Kruskal-Wallis test followed by Dunn´s multiple comparison was applied to the dataset. **B** WT and *Ffar2/Ffar3*[-/-] mice were transplanted with human microbiota from ABX-naïve or antibiotic-treated patients, IMs were isolated from the bone marrow by FACS sorting, incubated with *K. pneumoniae*, and CFUs were counted at the indicated time points. Values are shown as median, each dot represents the mean data from triplicates from a single mouse (*n* = 4 for *K. pneumoniae* without IMs; *n* = 6 for IM[WT ØABX]; *n* = 4 for IM[WTABX]; *n* = 5 for IM[*Ffar2/Ffar3-/- ØABX*]; *n* = 5 for IM[*Ffar2/Ffar3-/- ABX*]). 2-way ANOVA test followed by Tukey´s multiple comparison between given time points and groups was applied to the datasets. *\*P* < 0.05, \*\**P* < 0.01, \*\*\**P* < 0.005.

and FACS-sorted for IMs. Sorted IMs were counted, washed twice with sterile 1 ml PBS and volume was adjusted to $6 \times 10^5$ IMs per 60 μl PBS. *Ccr2*[-/-] mice were immediately injected intravenously via tail vein with PBS containing $6 \times 10^5$ WT or *Ffar2/Ffar3*[-/-] IMs or were injected with sterile PBS containing no cells as controls. *Ccr2*[-/-] mice were infected 30 minutes after adoptive cell transfer. IM transfer were checked 24 h post infection and transfer via FACS.

### Fecal DNA extraction and shotgun sequencing

Human and mouse fecal samples were collected and stored at −80 °C. DNA was isolated using the ZymoBIOMICS™ DNA Miniprep Kit (Zymo Research, Cat#D4300). DNA concentrations and purities were quantified by spectrophotometry using a NanoDrop™ 2000 (Thermo-Fisher). Metagenomic shotgun sequencing was performed by LGC Genomics GmbH (Berlin, Germany) and Eurofins (Ebersberg, Germany) on the Illumina NovaSeq platform.

### Metagenome bioinformatics analysis

Sequences retrieved from metagenomic shotgun sequencing were processed with the NGless pipeline (v1.3)[43]. Reads were trimmed with a Phred-score <25 and discarded if the sequence length was below 45 bp. Contamination filtering was performed by mapping the reads against GRCh38.p14 for human-derived stool samples and with GRCm39 for mouse-derived stool samples, respectively. Both references have been masked by mapping bacterial genes from the proGenomes2 database[44]. Taxonomic profiling was conducted using the NGless mOTUs module (v2.6). Functional profiling was performed by

mapping reads to the Global Microbial Gene Catalog (GMGC, human-gut (v1))[45] and further aggregated at the KEGG KO level. Samples with fewer than 1000 reads were excluded from the analysis. Species detected in at least 10% of the samples with an overall study-wide abundance of at least 10e-4 were retained for further analysis. Species level alpha and beta diversity were calculated using *vegan (v2.6-4)* package (Shannon index) and the *stats (v4.2.2)* package. The SCFA-pathways (acetate, butyrate, propionate) were binned manually from KEGG KOs (see supplementary data 2). *Orddom (v3.1)* package was used to calculate Cliff's delta, as a measure of standardized effect sizes, of microbial species abundance and SCFA-pathway abundances across the different group comparisons. As a focused hypothesis testing, only the species that were found to be differently regulated between humans receiving antibiotics vs not receiving, were specifically tested in mice. FDR correction was done to account for multiple testing using the Benjamini-Hochberg method. All microbiome analysis was carried out in the R statistical programming language (*v4.2.2*) and figures were generated using *ggplot2 (v3.4.0)*.

### single cell RNA sequencing

WT- and *Ffar2/Ffar3*[-/-] mice were euthanized 12 h after infection or PBS treatment. Lungs were perfused with 10 mL cold PBS followed by 2 mL dispase (Corning). Afterwards a BAL canula was inserted into the trachea, lungs were filled with 700 μl dispase and 500 μl low melting agarose (Invitrogen), and incubated for 5 min. Lungs were then removed, dipped in ice cold PBS and afterwards minced with tweezers in a petri dish and incubated in DNAse/collagenase for 30 min at 37 °C with an agitation speed of 125 rpm. Cell suspensions were centrifuged for 5 min at 4° and 1,300 g, supernatants were discarded, red blood cell lysis was applied, cells were counted, and dead cells were removed using a dead cell removal kit (Stemcell Technologies). Cell counting and hashtagging was performed using TotalSeq anti-mouse hashtag antibodies purchased from BioLegend. Single-cell suspensions of PBS-treated WT animals were stained with hashtag 1 (clone M1/42, #155801), PBS-treated *Ffar2/Ffar3*[-/-] mice with hashtag antibody 2 (clone M1/42, #155803), infected WT mice with hashtag 3 (clone M1/42, #155805), and infected *Ffar2/Ffar3*[-/-] animals with hashtag 4 (clone M1/42, #155807). The hashtagged single-cell suspensions were counted, pooled to 50,000 cells per group, and libraries were generated using Chromium Single Cell 5´ (v2) Reagent Kit (10x Genomics). The generated libraries were sequenced on a NovaSeq 6000 S4 flowcell type with 200 cycles with 30,000 cells per lane and four lanes in total, targeting 2500 million reads/ lane.

### scRNAseq data analysis

Raw sequencing data was de-multiplexed and quality-checked using the Cell Ranger pipeline. In brief, BCL files from each library were converted to FASTQ reads using bcl2fastq Conversion Software (Illumina) using the respective sample sheet with the 10x barcodes and TotalSeq antibodies utilized. Then, the reads were further aligned to the reference genome provided by 10x Genomics (Mouse reference mm10) and a digital gene expression matrix was generated to record the number of UMIs for each gene in each cell. Count matrices were then loaded in R (version 4.2.1) and quality metrics were applied. Thresholds were chosen as followed: Number of genes (>150 and <4000), percentage of mitochondrial reads (<10%). Standard pipeline of Seurat package[46] (version 4.2.0) was used to integrate datasets (IntegrateData), finding variable features (FindVariableFeatures), scaling data (ScaleData) and calculating PCA based on highly variable genes (RunPCA). If applied, batch correction was conducted via Harmony package[47]. Furthermore, RunUMAP, FindNeighbours and FindClusters functions were applied. To calculate differential gene expression, the Seurat function FindMarkers was used. Graphs were generated with the DimPlot, DotPlot and DoHeatmap. GSEA analysis was performed utilizing the fgsea package in R[48]. Volcano plots were

generated using ggplot2 package and data wrangling was performed by dplyr and tidyr package.

## SCFA measurement

Human plasma was extracted and measured as recently described[49]. Briefly, 40 μL of plasma was mixed with 20 μL of 200 mM 3NPH (3-nitrophenylhydrazine hydrochloride), 20 μL of 120 mM EDC (N-(3-dimethylaminopropyl)-N'-ethylcarbodiimide hydrochloride) with 6% pyridine, and 10 μL of 50 μM internal standard. The mixture was incubated at 40 °C for 30 min. Afterwards, 410 mL of 10% (v/v) ACE:$H_2O$ (acetonitrile:water) was added following centrifugation at 5500xg for 20 min at room temperature. The supernatants were transferred into new vials and analyzed by UHPLC (ultra high performance liquid chromatography) system 1290 Infinity II coupled to a TSQ Quantiva. 5 μL of sample was injected and chromatographic separation was achived with an RP C18 column AQUITY BEH 1.7 μm 2.1 × 100 mm at 40 °C in a gradient of water ( = solvent A with 0.1% formic acid) and acetonitrile ( = solvent B with 0.1% formic acid).

## Statistical analysis

GraphPad Prism 9 software was used for graphical visualization and statistical analysis. The specific statistical tests used are indicated in the figure legends. For all statistical analyses, $P$ values < 0.05 were considered significant.

## Reporting summary

Further information on research design is available in the Nature Portfolio Reporting Summary linked to this article.

## Data availability

Source data are provided with this paper. All data generated in this study (except the sequencing data) have been deposited in the Figshare depository (https://doi.org/10.6084/m9.figshare.25243333). The raw microbiome shotgun sequencing and single-cell RNA sequencing data generated in this study have been deposited in the NIH Sequence Read Archive under accession code BioProject ID PRJNA1064467. Source data are provided with this paper.

## Code availability

The R code used for statistical analysis and to generate the figures in this study is archived in Zenodo with the https://doi.org/10.5281/zenodo.10526874.

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

## Acknowledgements

The authors are grateful to all patients for consenting to biosampling and data collection. We would like to thank Ulrike Fiebiger, and the staff of the animal research facility of the Charité – Universitätsmedizin Berlin for excellent technical assistance, animal breeding and generation of secondary abiotic mice. This work was supported by the German Research Foundation (SFB-TR84 A5 to B.O. and M.M.H., OP 86/12-1 to B.O., OP 86/13-1 to B.O., and SFB 1449 B02 to M.W.), the German Federal Ministry of Education and Research (MAPVAP FKZ 01KI2124 to B.O. and M.W., and JPI AMR-EMBARK F01KI1909A to U.L. and S.K.F.), the European Research Council under the European Union's Horizon 2020 research and innovation program (grant 852796 to N.W.), the Corona Foundation in the German Stifterverband (to N.W.), the HMWK LOEWE program (Diffusible Signals projekt D2 to L.N.S.).

## Author contributions

Conceptualization: B.O. Patient samples collection: B.M.P.L., M.K., V.I., F.K., L.P., J.S. Experimental tools: S.O., S.J., Ma.M., M.W. Investigation: P.J.D., I.R., F.F.V., M.H., G.A.H., M.F.M., A.B.K., S.C., S.M.W., B.G. Analyses: P.J.D., H.A., I.R., Mi.M., U.L., H.B., L.N.S., D.B., N.W., U.B., R.F.G., J.A.K, and S.K.F. Funding acquisition: B.O., M.M.H. Project administration: B.O., M.M.H. Supervision: B.O., M.M.H. Writing – original draft: B.O., P.J.D., H.A., N.W. Writing – review & editing: P.J.D., B.O. and all other authors.

## Funding

## Competing interests

The authors declare no competing interests.

## Additional information

¹Department of Infectious Diseases, Respiratory Medicine and Critical Care, Charité – Universitätsmedizin Berlin, Corporate Member of Freie Universität Berlin
and Humboldt-Universität zu Berlin, Berlin, Germany. ²Experimental and Clinical Research Center, a cooperation of Charité – Universitätsmedizin Berlin and
Max-Delbrück-Center for Molecular Medicine, Berlin, Germany. ³Max-Delbrück-Center for Molecular Medicine in the Helmholtz Association, Berlin, Germany.
⁴DZHK (German Centre for Cardiovascular Research), partner site Berlin, Berlin, Germany. ⁵Department of Nephrology and Internal Intensive Care Medicine,
Charité – Universitätsmedizin Berlin, Corporate Member of Freie Universität Berlin and Humboldt-Universität zu Berlin, Berlin, Germany. ⁶Metabolomics
Platform, Berlin Institute of Health at Charité, Berlin, Germany. ⁷Core Unit Bioinformatics, Berlin Institute of Health at Charité, Berlin, Germany. ⁸German
Rheumatism Research Center, a Leibniz Institute, Berlin, Germany. ⁹Department of Medicine, Institute for Lung Research, Philipps University Marburg,
Marburg, Germany. ¹⁰German center for lung research (DZL), Marburg, Germany. ¹¹Max-Planck-Institute for Heart and Lung Research, Bad Nauheim, Germany.
¹²Department of Nephrology, University Hospital Regensburg, Regensburg, Germany. ¹³German center for lung research (DZL), Berlin, Germany. ¹⁴Institute of
Microbiology, Infectious Diseases and Immunology, Charité – Universitätsmedizin Berlin, Corporate Member of Freie Universität Berlin and Humboldt-
Universität zu Berlin, Berlin, Germany. ¹⁵Structural and Computational Biology Unit, European Molecular Biology Laboratory (EMBL), Heidelberg, Germany.
✉e-mail: bastian.opitz@charite.de

