## [Peer Review File · Nature Communications]

REVIEWER COMMENTS

Reviewer #1 (Remarks to the Author):

This study, by Dorner et al, aims to understand how disruption of the human intestinal microbiota by antibiotic therapy disturbs pulmonary defenses against pneumonia caused by the major human pathogen *Klebsiella pneumoniae*. The authors use a range of tools to undertake this task: they have a bank of human microbiota samples and have sequenced them; they have mouse models of pneumonia and ex vivo assays of innate immune cell antimicrobial activity. Their major conclusion from their study is that there are reduced numbers of commensals in the human microbiota that can produce SCFAs and this is what drives the increase in susceptibility to pneumonia in patients receiving antibiotics. The study is potentially interesting, I have the following comments on the study:

- 1) The major narrative of the paper is that reduced SCFA production by the microbiota of patients receiving antibiotics is the main driver of reduced pulmonary immunity. Unless I missed the data, I cannot see any measurement of fecal SCFA concentration. For this narrative to be substantiated SCFA levels need to be biochemically quantified, relying on sequencing data to infer metabolite levels is not sufficiently robust.
- 2) SCFA have effects on the host that are not mediated by FFARs, they can go through their ability to inhibit HDACs. Some demonstration in the authors models that restoration of SCFAs in the FFAR KO mice doesn't rescue defects in pulmonary defenses would strengthen the argument the effects of the microbiota are going via FFARs.
- 3) How closely are the comparisons between FFAR KO v WT mice mimicking the WT mice + antibiotics v WT mice – antibiotics? For example, in the supplementary data (SFig1), the authors do an extensive analysis of the cytokine response in the lungs of FFAR KO v WT mice and also, though separately, WT + ABX v WT - ABX but despite analyzing approximately 30 cytokines the overlap between the antibiotic comparisons and the FFAR comparisons is only about 4 cytokines. Thus, it is unclear whether the inflammatory response controlled by the microbiota (as determined in the +/- ABX experiments) is similar to the how SCFA are controlling the inflammatory response in the lung (as determined in the +/- FFARs)?
- 4) Can the authors clarify and provide justification of their experiments transferring the human microbiota to mice. If I understand correctly, they used 29 stool samples and basically gave one WT mouse a single microbiota sample and one FFAR KO mouse a single stool sample. This pattern was then repeated with pairs of mice and a different stool sample etc. It seems to me that in aggregate this is an impressive number of different stool samples to test, however, each one is only analyzed in a tiny number of mice. Robustness is therefore a potential issue, are the effects of the microbiota reproducible and what is the statistical power justification?
- 5) Is the depletion of neutrophils shown in in SFig4 statistically significant?

Reviewer #2 (Remarks to the Author):

In this study, Dörner et al investigated how Clinically used broad-spectrum antibiotics affect inflammatory monocyte-dependent antibacterial defense in lung. They found that antibiotic therapy in hospitalized patients is associated with dysbiosis and decreased short-chain fatty acid producing bacteria, and these antibiotic-induced microbiota compromise pulmonary defense against MDR *Klebsiella pneumoniae* by inhibiting (FFAR)2/3-controlled antibacterial activity of inflammatory monocytes. Although this study provides some interesting insights, it is rather preliminary at the current form. Most of the conclusions are based on association but lacking solid mechanistic data. It will be helpful if a study demonstrating how the altered gut microbiota inhibits FFAR2/3 expression and how SCFA and FFAR2/3 regulate PMN and inflammatory monocyte. Some other concerns need to be addressed.

- 1) Fig 1 E and F. It is hard to read. please make the labels clearly indicating what they mean.
- 2) Fig 4 F/I/J. Any of those differentially expressed genes or pathways play a role in SCFA/FFAR2/3 regulation of PMN and inflammatory monocyte antimicrobial activity?
- 3) Fig 5A/B/C. Those data can not define IMs mediate FFAR2/3-dependent pulmonary antibacterial defense as the authors claimed as they were only association studies, which is the key point of this study. IM specific FFAR2/3 KO mice are needed to make such conclusion.

Reviewer #3 (Remarks to the Author):

This is an interesting and relevant study showing that antibiotic-induced changes in the human intestinal microbiota can impair defences against bacterial infections in the lung through the effect of SCFA on the activity of inflammatory monocytes. Previous studies have already known that the intestinal microbiota can affect lung immunity, and that immune cell activity can be modulated by SCFAs produced by the intestinal microbiota. The novelty of this study resides in (i) the utilization of a fecal-transplantation mouse model, in combination with human fecal samples characterised from antibiotic-treated patients, to show that changes in the microbiota from patients can have an impact on the defence against *K. pneumoniae* infection, (ii) the identification of a novel mechanism by which lung defences are impaired by microbiota depletion (i.e. effect of SCFA in the activity of inflammatory monocytes). Overall the manuscript is well written, the experiments well designed and the conclusions are in general supported by the data. Nevertheless, additional analysis, as specified below, could be performed to strength the conclusions of this manuscript:

- Levels of SCFAs could be quantified both for the human and mouse data. Although shotgun sequencing results suggests that SCFA production is impaired in antibiotic-treated patients and mice, considering the relevance of SCFA in this study, it will be important to directly measure the levels of these metabolites in fecal human and murine samples.
- The differences in the levels of *K. pneumoniae* of mice transplanted with the human microbiota from antibiotic-treated patients vs naive is clearly significantly different. But it will be good to show the clinical relevance of this difference. Is lung inflammation or mouse survival different in these two groups of mice?

- It will be interesting to decipher if the effect of SCFA on the activity of IMs is direct or indirect. A simple experiment based on Fig. 5C could be to obtain IMs from microbiota depleted mice and incubate them with SCFAs before co-incubation with *K. pneumoniae*. Would the addition of SCFAs boost the activity of IMs against *K. pneumoniae*? A similar experiment could be performed with IMs from FFAR KO mice as control.

- In the line of the previous question:

- (i) Can IMs from WT mice rescue the lower response against Kp found in KO mice?

- (ii) Could the effect of lacking the FFAR on the studied phenotype be indirect (i.e. through differences in the microbiota in these two groups of mice that could impact response to *k.pneumoniae*)

- Some of the changes in the expression detected in the KO mouse strain using single cell sequencing could be confirmed through qPCR. In addition, using this approach, it could be verified that similar changes in the expression are found in mice treated with antibiotics that received the naive-microbiota vs antibiotic-microbiota from patients or in mice treated with antibiotics vs conventionalized mice.

Other comments:

- Authors could have labeled the numbers of each line and pages to facilitate the reviewer response.

- Patients characteristics: can the authors perform statistical tests to analyze that there are no differences (besides antibiotic treatment) between both groups of patients.

- What is the reason of not performing shotgun metagenomic sequencing in all the patients and just a subset of 55 patients?

- Fig. 1E: to better understand the changes detected and to know in how many patients occur these changes, a heatmap could be shown including the data from each patient. Alternatively, bargraphs including one dot per sample could also clarify the changes induced by antibiotics in each patient. This could also be done for Fig. 2. and a similar approach could be used to show more clearly changes observed in SCFA modules both in Fig 1 and Fig 2.

- Fig. 2: How many of the significant changes detected in humans could be detected in mice? Can the authors specify in the text the % of changes that could be recapitulate in their mouse model?

- Can the authors show the number of human bacterial species that were able to colonize mice? Also how many of the total microbiota were bacterial species derived from mice? Upon antibiotic cessation there can be some degree of murine microbiota recovery.

- Can the authors show the level of depletion obtained with the antibiotic treatment?

- Fig. 2D-E. Albeit not statistically significant, FFAR KO mice that received the antibiotic-microbiota have higher levels of *K. pneumoniae* than those receiving the naive microbiota. The authors should discuss that other mechanisms dependent on the microbiota composition but independent on the FFAR receptor could be playing a role in responses to *K.pneumoniae* infection.

- Fig 3A. Why there is a difference between ABX mice and ABX FFAR KO mice in the levels of *K. pneumoniae*? Since ABX treated mice should contain very little levels of SCFA, one should not expect a difference between these two groups of mice? Could the FFAR receptor have a role in the development of specific immune populations and this defect may not be recapitulated by transient reduction of SCFA levels?

- "We did not observe any significant impact of FFAR2/FFAR3 deficiency on cytokines" This is true except for IL-10 that it is slightly significantly lower in KO mice.

- "nor did we observe an effect of microbiota depletion (Fig. S1C)": I cannot find this result.

- Fig. 4: "possibly partly due to the lower cell numbers analyzed"
- how many cell numbers were analyzed in each case?
- Suppl. Fig. 4A. If the data is available, it will be nice to show that CSf2 KO mice have similar levels of IMs and PMNs and therefore the depletion is specific for AMs. A similar analysis could be done for Suppl. Fig 4B and Suppl. Fig. 4C.
- Could you eliminate the last part of the statistical analysis since it seems to be instructions from the manuscript?

Reviewer: 1

We would like to thank the reviewer for his/her time and effort spent on evaluating our manuscript and for the positive reception of our manuscript.

1) The major narrative of the paper is that reduced SCFA production by the microbiota of patients receiving antibiotics is the main driver of reduced pulmonary immunity. Unless I missed the data, I cannot see any measurement of fecal SCFA concentration. For this narrative to be substantiated SCFA levels need to be biochemically quantified, relying on sequencing data to infer metabolite levels is not sufficiently robust.

Response: We agree with the reviewer that measuring SCFA levels is important to support our conclusion. Since SCFA likely influence IMs through the circulation, we consider their plasma level as even more interesting in the context of our study. In addition, we did not have enough stool samples left from the patients who had donated stool samples for sequencing and functional analysis, but we did have plasma. We therefore quantified SCFA in plasma samples from patients receiving antibiotics and patients not receiving antibiotics. Largely in line with the narrative of our study, we found that several SCFAs were reduced in plasma of patients receiving antibiotics (see new Fig. 1F-J).

2) SCFA have effects on the host that are not mediated by FFARs, they can go through their ability to inhibit HDACs. Some demonstration in the author's models that restoration of SCFAs in the FFAR KO mice doesn't rescue defects in pulmonary defenses would strengthen the argument the effects of the microbiota are going via FFARs.

Response: Following the reviewer's helpful suggestion, we have performed additional in vivo experiments. The new Fig. 3C show that SCFA act via FFAR2/FFAR3 in vivo. In addition, we demonstrate in the new Fig. 6A that SCFA directly influence the antibacterial activity of inflammatory monocytes via FFAR2/FFAR3.

3) How closely are the comparisons between FFAR KO v WT mice mimicking the WT mice + antibiotics v WT mice – antibiotics? For example, in the supplementary data (SFig1), the authors do an extensive analysis of the cytokine response in the lungs of FFAR KO v WT mice and also, though separately, WT + ABX v WT - ABX but despite analyzing approximately 30 cytokines the overlap between the antibiotic comparisons and the FFAR comparisons is only about 4 cytokines. Thus, it is unclear whether the inflammatory response controlled by the microbiota (as determined in the +/- ABX experiments) is similar to the how SCFA are controlling the inflammatory response in the lung (as determined in the +/- FFARs)?

Response: We have indeed analysed cytokine production in WT mice with or without antibiotic treatment (Fig. S3C) as well as in WT and *Ffar2/Ffar3*^{-/-} animals (Fig. 3D, S3B). The overlap between the "antibiotic comparisons" and the "FFAR comparisons" was however not only about 4 but actually 9 cytokines. In fact, TNF α , IL-1 β , IL-6, IL-10, CXCL1, CXCL2, CCL2, G-CSF, GM-CSF have been measured in both experiments, and all of these cytokines except IL-10 were neither affected by microbiota depletion ("antibiotic comparisons") nor by FFAR2/FFAR3 deficiency ("FFAR comparisons"). The only exception is IL-10 whose production was slightly reduced in *Ffar2/Ffar3*^{-/-} animals but apparently not in microbiota-depleted animals.

However, considering that reduced production of the anti-inflammatory IL-10 is unlikely to be responsible for the enhanced infection susceptibility of *Ffar2/Ffar3*^{-/-} animals, and that the reduction of IL-10 production was only mild, we do not consider this difference as biologically important in the context of our study.

4) Can the authors clarify and provide justification of their experiments transferring the human microbiota to mice. If I understand correctly, they used 29 stool samples and basically gave one WT mouse a single microbiota sample and one FFAR KO mouse a single stool sample. This pattern was then repeated with pairs of mice and a different stool sample etc. It seems to me that in aggregate this is an impressive number of different stool samples to test, however, each one is only analyzed in a tiny number of mice. Robustness is therefore a potential issue, are the effects of the microbiota reproducible and what is the statistical power justification?

Response: We apologize for the typo, in fact 24 different patient stool samples were used to transplant 48 animals. In most cases, a pair of WT and *Ffar2/Ffar3*^{-/-} mice received the same patient fecal sample. Since some animals died before the infection, some pairs were affected by this drop-out. Since the inter-individual microbiota variability in humans is usually high, we considered a sufficient number of human donors as adequate and necessary to compare the effect of antibiotic-naïve vs. antibiotic-associated microbiota. Our approach is also largely in line with the recently published recommendations for the use of human microbiota-associated mice (Arrieta et al. Human Microbiota-Associated Mice: A Model with Challenges. *Cell Host Microbe*. 2016;19(5):575-8). Since we hypothesized that the antibiotic treatment leads to an impaired bacterial clearance we treated all samples from patients that received antibiotics as one group. Prior experience with our model suggested that the within-sample variance (of mice recolonized with the same sample) could be expected to be smaller than the variance between samples of the same group (patients treated with antibiotics vs. controls).

For the statistical power and effect size justification for this experiment, we used the means of bacterial burdens in conventionally housed and microbiota-depleted WT of 12 mice per group. The means for this groups were 1.22×10^7 for group 1 (WT conv.), 3.69×10^7 for group 2 (WT ABX), common SD of 2.1×10^7 . With a group size of 2, a significance niveau of $\alpha = 0.05$ und and a power of $1 - \beta = 0.8$ the calculated effect size was 0.47. In our hypothesis, we were expecting a similar effect size of 0.65 in our microbiota transplanted mice experiment. With the calculated effect size of 0.65 in our translational trial, we expected a total sample size of $n=48$ ($n=12$ mice per group) for this experiment to find statistical differences. In total, we conducted the experiment with 12 mice per group, but for some groups, the numbers of mice are slightly lower since a few mice did not survive the stool transplantations or died before infection with *K. pneumoniae*.

5) Is the depletion of neutrophils shown in in Fig. S4 statistically significant?

Response: The neutrophil depletion is now significant (see revised figures 5B and S6E)

Reviewer: 2

We would like to thank the reviewer for his/her time and effort spent on evaluating our manuscript.

“It will be helpful if a study demonstrating how the altered gut microbiota inhibits FFAR2/3 expression and how SCFA and FFAR2/3 regulate PMN and inflammatory monocyte.”

Response: There seems to be a misunderstanding here: The microbiota influences inflammatory monocyte-dependent antibacterial defense through SCFAs and their receptors FFAR2/3, but does not inhibit FFAR2/3 expression. The new Fig. 1E shows that the antibiotic-induced microbiota alterations are indeed associated with differences in levels of several SCFAs, indicating that microbiota perturbations influence SCFA production. These microbiota alterations and altered SCFA production influences the antimicrobial activity of inflammatory monocytes via FFAR2/3 (see new Fig. 6A and Fig. 6B). The SCFA-induced enhancement of the antibacterial activity of inflammatory monocytes is further dependent on phagocytosis, endolysosomal acidification and lysozyme activity, as indicated by the new data demonstrating that acetate-induced antibacterial activity of inflammatory monocytes is blocked by cytochalasin D, bafilomycin A1, and N,N',N''-Triacetylchitotriose (see new Fig. 6A). PMNs do not seem to play a role in the FFAR2/FFAR3-dependent strengthening of antibacterial defense against *K. pneumoniae* (Fig. 5B).

1) Fig 1 E and F. It is hard to read. Please make the labels clearly indicating what they mean.

Response: We apologize for this shortcoming. The figures 1E (now Fig. 1D) and F (now Fig. 1E) and their labelling have been revised.

2) Fig 4 F/I/J. Any of those differentially expressed genes or pathways play a role in SCFA/FFAR2/3 regulation of PMN and inflammatory monocyte antimicrobial activity?

Response: Our new Fig. 6A indicates that the SCFA-induced enhancement of the antibacterial activity of inflammatory monocytes is dependent on phagocytosis, endolysosomal acidification and lysozyme activity. This is indicated by our data showing that cytochalasin D, bafilomycin A1, and N,N',N''-Triacetylchitotriose block the SCFA-induced antibacterial activity of inflammatory monocytes. These results are thus in line with the transcriptome data, which indicate that SCFA/FFAR2/3 influence expression of the genes related to e.g. endocytosis, lysosomes, and lytic vacuole in inflammatory monocytes (Fig. 4I, J). In future studies, we aim to further investigate the molecular mechanisms of how SCFA/FFAR2/3 enhance IM's antibacterial activity. However, such studies are complex and require various additional in vitro and in vivo experiments, which is beyond the scope of our current study.

3) Fig 5A/B/C. Those data can not define IMs mediate FFAR2/3-dependent pulmonary antibacterial defense as the authors claimed as they were only association studies, which is the key point of this study. IM specific FFAR2/3 KO mice are needed to make such conclusion.

Response: We appreciate the reviewers' comment and agree that further evidence was required to support our conclusion. However, we are not aware of any IM-specific Cre-expressing driver mouse line that could be used to generate IM-specific *Ffar2/Ffar3*^{-/-} animals. We therefore choose a slightly different approach to further prove our conclusion that IMs mediate FFAR2/3-dependent pulmonary antibacterial defense. We adoptively transferred IMs from WT or *Ffar2/Ffar3*^{-/-} mice into *Ccr2*^{-/-} animals, which largely lack the ability to recruit their own IMs to the lungs. We observed that *Ccr2*^{-/-} mice receiving WT IMs had lower bacterial loads as compared to *Ccr2*^{-/-} mice receiving *Ffar2/Ffar3*^{-/-} IMs or PBS (see new Fig. 5D), further supporting our conclusion that IMs contribute critically to the FFAR2/3-dependent defense augmentation against *K. pneumoniae*. Moreover, our new Fig. 6A shows that SCFA directly enhance the antibacterial activity of IMs via FFAR2/FFAR3, and Fig. 5C provides strong evidence for FFAR2/FFAR3 and IMs acting in the same pathway.

Reviewer: 3

We would like to thank the reviewer for his/her time and effort spent on evaluating our manuscript and for the positive reception of our manuscript.

- Levels of SCFAs could be quantified both for the human and mouse data. Although shotgun sequencing results suggests that SCFA production is impaired in antibiotic-treated patients and mice, considering the relevance of SCFA in this study, it will be important to directly measure the levels of these metabolites in fecal human and murine samples.

Response: We agree with the reviewer that measuring SCFA levels is important to support our conclusion. Since SCFA likely influence IMs through the circulation, we consider their plasma level as even more interesting in the context of our study. Moreover, unfortunately we did not have enough stool samples left from the patients (who had donated stool samples for sequencing and functional analysis) or from mice transplanted with human fecal material), but we did have plasma from patients. We therefore quantified SCFA in plasma samples from patients receiving antibiotics and patients not receiving antibiotics. Largely in line with the narrative of our study, we found that several SCFAs were reduced in plasma of patients receiving antibiotics (see new Fig. 1F-J).

- The differences in the levels of *K. pneumoniae* of mice transplanted with the human microbiota from antibiotic-treated patients vs naive is clearly significantly different. But it will be good to show the clinical relevance of this difference. Is lung inflammation or mouse survival different in these two groups of mice?

Response: We thank the reviewer for this helpful comment. According to his/her suggestion, we have analysed MPO as a marker for neutrophilic inflammation and serum albumin as a marker for lung barrier failure in BALF of WT mice transplanted with antibiotic-naïve or antibiotic-associated patient microbiota. Our new data reveal that after infection with *K. pneumoniae*, mice transplanted with antibiotic-associated microbiota show enhanced inflammation and increased barrier dysfunction as compared to mice harboring an antibiotic-naïve patient microbiota (see new Fig. 2F, G).

- It will be interesting to decipher if the effect of SCFA on the activity of IMs is direct or indirect. A simple experiment based on Fig. 5C could be to obtain IMs from microbiota-depleted mice and incubate them with SCFAs before co-incubation with *K. pneumoniae*. Would the addition of SCFAs boost the activity of IMs against *K. pneumoniae*? A similar experiment could be performed with IMs from FFAR KO mice as control.

Response: Our new data show that SCFA directly enhance the antimicrobial activity of inflammatory monocytes, and that this effect is dependent on FFAR2/3 (see new Fig. 6A).

- Can IMs from WT mice rescue the lower response against Kp found in KO mice?

Response: Largely in line the reviewer's advice, we performed an additional experiment to further prove that IMs are responsible for the FFAR2/3-mediated effect on antibacterial immunity. We chose a model in which we adoptively transferred WT or *Ffar2/Ffar3*^{-/-} IMs into *Ccr2*^{-/-} animals (that are deficient in the ability to recruit their own IMs to the lungs). We choose this experimental setup as we were aiming for a model in which little endogenous IMs were potentially interfering with the adoptively transferred cells in the lung tissue. We observed that *Ccr2*^{-/-} mice transplanted with WT IMs had lower bacterial loads as compared to *Ccr2*^{-/-} mice receiving PBS or *Ffar2/Ffar3*^{-/-} IMs (see new Fig. 5D). These new data further support our conclusion that IMs contribute critically to the SCFA/FFAR2/3-dependent defense augmentation against *K. pneumoniae*.

- Could the effect of lacking the FFAR on the studied phenotype be indirect (i.e. through differences in the microbiota in these two groups of mice that could impact response to *K. pneumoniae*)

Response: We consider this possibility as extremely unlikely for several reasons: First, in our human patient microbiota transfer experiments, we transplanted in most cases one human fecal sample into a pair of WT and *Ffar2/Ffar3*^{-/-} animals, which were subsequently co-housed for the rest of the experiment. Any potential FFAR2/FFAR3-dependent shift in the microbiota composition would have likely been transferred between WT and *Ffar2/Ffar3*^{-/-} animals as mice are coprophagic. Second, our human microbiota transplantation experiments demonstrate that antibacterial resistance of transplanted WT mice (and inflammatory monocytes) correlates with the capacity of the human microbiota to produce SCFA. Third, antimicrobial defense is not only reduced in *Ffar2/Ffar3*^{-/-} animals but also increased in mice treated with SCFA.

- Some of the changes in the expression detected in the KO mouse strain using single cell sequencing could be confirmed through qPCR. In addition, using this approach, it could be verified that similar changes in the expression are found in mice treated with antibiotics that received the naive-microbiota vs antibiotic-microbiota from patients or in mice treated with antibiotics vs conventionalized mice.

Response: Understanding the molecular mechanisms underlying the SCFA-dependent enhancement of antibacterial defenses of IMs is another interesting point. In a future study, we indeed plan to use scRNAseq and bulkRNA seq to investigate the bone marrow cells and in particular IMs from WT and *Ffar2/Ffar3*^{-/-} animals transplanted with different human microbiotas. However, these investigations are complex, time-consuming and require a new study to recruit respective patients and collect their fecal samples, as unfortunately our previous stool samples are largely exhausted. However, we have instead started to perform some functional studies on IMs from WT and *Ffar2/Ffar3*^{-/-} mice, which show that SCFA-induced enhancement of the antibacterial activity of IMs is dependent on phagocytosis, endolysosomal acidification and lysozyme activity (new Fig. 6A). These functional data are potentially in line with our transcriptome data (shown in Fig. 4I and J), indicating that genes related to e.g. endocytosis, lysosomes, and lytic vacuole are affected.

Other comments:

- Authors could have labeled the numbers of each line and pages to facilitate the reviewer response.

Response: We apologize for this shortcoming. Line and page numbers have now been added to the manuscript.

- Patients characteristics: can the authors perform statistical tests to analyze that there are no differences (besides antibiotic treatment) between both groups of patients.

Response: Following the reviewers advice, we have performed such statistical tests (please see revised table 1)

- What is the reason of not performing shotgun metagenomic sequencing in all the patients and just a subset of 55 patients?

Response: We collected fecal samples from 72 hospitalized patients receiving antibiotics or not. Due to technical problems with the DNA isolation or sequencing of some of the samples (low DNA yield, low sequencing depth), we successfully sequenced only 55 of these samples.

- Fig. 1E: to better understand the changes detected and to know in how many patients occur these changes, a heatmap could be shown including the data from each patient. Alternatively, bargraphs including one dot per sample could also clarify the changes induce by antibiotics in each patient. This could also be done for Fig. 2. and a similar approach could be used to show more clearly changes observed in SCFA modules both in Fig 1 and Fig 2.

Response: We thank the reviewer for the valuable suggestion. We now show the significantly altered species in patients and SCFA modules in the revised Fig. 1D and E as heatmaps. In Fig. 2, we would prefer the previous form of presentation, as it makes it easier to show similarities between the human donors (fecal donors) and the mice (fecal recipients).

- Fig. 2: How many of the significant changes detected in humans could be detected in mice? Can the authors specify in the text the % of changes that could be recapitulate in their mouse model?

Response: In line with the general efficacy of the fecal material transfer (FMT), we can detect 89% (49/58) significantly dysregulated species from the human cohort in our FMT mouse model. The key limitation of comparing species is the high interindividual difference in human microbiome composition. This is exaggerated by a shift of taxonomic composition upon transfer into the mouse model. We found a high overlap of 60% (29/49) altered species with similar directionality of effect sizes (e.g. up in antibiotics-native feces or down in antibiotic-native feces in mice and human). However, we only reach statistical significance in 3 of those (10%). This is most likely due to a lack of statistical power. However, on a functional level we can capture the same difference as in the human cohort in our FMT mouse model (e.g. depletion of specific taxa and SCFA modules, see Fig. 2B, C). This indicates that a key feature of the microbiome (functional capacity) is truthfully transferred.

- Can the authors show the number of human bacterial species that were able to colonize mice? Also how many of the total microbiota were bacterial species derived from mice? Upon antibiotic cessation there can be some degree of murine microbiota recovery.

Response: We successfully transferred 145 ± 28 (mean \pm SD) species from human to mouse. We found no difference in the effectiveness of the transfer between antibiotic-naïve and antibiotic fecal samples (see new Fig. S2A). The transferred bacterial species made up 80-90% of the donor microbiota and 75-90% of the mouse microbiota after transfer (see new Fig. S2B). Again, we found no statistically significant group effect.

- Can the authors show the level of depletion obtained with the antibiotic treatment?

Response: To confirm successful depletion of the murine commensal gut microbiota, we applied cultural analyses of fecal samples derived from antibiotic-treated mice. Direct plating and enrichment procedures revealed that all fecal samples were culture-negative for aerobic, microaerobic and obligate anaerobic bacterial species (not shown). Previously, we additionally assess abundance of main bacterial groups by quantitative 16S rRNA-based PCR analysis of fecal samples derived from conventionally colonized and antibiotic-treated mice as compared to autoclaved food pellets (Ref. 1 and Fig. I below). In antibiotic-treated mice, bacterial 16S rRNA gene numbers were decreased by up to 10 orders of magnitude compared to SPF mice. Remarkably, 16S rRNA gene numbers in fecal samples from antibiotic-treated mice and in autoclaved food pellets were comparable indicating a successful and biologically relevant depletion of the murine gut microbiota following antibiotic-treated treatment.

Fig. I: Intestinal microbiota composition of conventional and secondary abiotic mice as compared to autoclaved food pellets. The intestinal microbiota composition was analysed in fecal samples derived from conventionally colonized (SPF) mice and mice subjected to an eight-week course of broad-spectrum antibiotic treatment (ABx) by quantitative Real-Time PCR amplifying variable regions of the bacterial 16S rRNA gene and compared to the bacterial composition detected in sterilized (autoclaved) food pellets. The following main intestinal bacterial groups were determined (expressed as 16S rRNA gene numbers per ng DNA): Enterobacteria (EB), enterococci (EC), lactic acid bacteria (LB), bifidobacteria (BIF), *Bacteroides/Prevotella* spp. (BP), *Clostridium coccooides* group (CLOCC), *Clostridium leptum* group (CLEP). Numbers of samples harboring the respective bacterial group out of the total number of analyzed samples are given in parentheses. Upon ABx treatment, the gene numbers measured in fecal pellets did not exceed those detected in autoclaved food samples.

(I) Ekmekciü I, von Klitzing E, Fiebigler U, Escher U, Neumann C, Bacher P, Scheffold A, Kühl AA, Bereswill S, Heimesaat MM. Immune Responses to Broad-Spectrum Antibiotic Treatment and Fecal Microbiota Transplantation in Mice. *Front Immunol.* 2017;8:397. DOI: 10.3389/fimmu.2017.00397.

- Fig. 2D-E. Albeit not statistically significant, FFAR KO mice that received the antibiotic-microbiota have higher levels of *K. pneumoniae* than those receiving the naive microbiota. The authors should discuss that other mechanisms dependent on the microbiota composition but independent on the FFAR receptor could be playing a role in responses to *K. pneumoniae* infection.

Response: We thank the reviewer for this helpful comment and have included a brief discussion about potential FFAR2/3-independent mechanisms (see page 8, lines 2-5 in the revised manuscript).

- Fig 3A. Why there is a differences between ABX mice and ABX FFAR KO mice in the levels of *K. pneumoniae*? Since ABX treated mice should contain very little levels of SCFA, one should not expect a difference between these two groups of mice? Could the FFAR receptor have a role in the development of specific immune populations and this defect may not be recapitulate by transient reduction of SCFA levels?

Response: We agree with the reviewer's interpretation of this small difference in the bacterial levels. It is well possible that FFAR2/FFAR3 have effects on immune cell development or imprinting that occurred before our mice were deprived of their microbiota. However, we also cannot exclude the possibility that FFAR2/FFAR3 are activated by a yet unknown additional endogenous ligand.

- "We did not observe any significant impact of FFAR2/FFAR3 deficiency on cytokines" This is true except for IL-10 that it is slightly significantly lower in KO mice.

Response: We apologize for not mentioning the small but significant difference in IL-10 levels. This is now corrected in the revised manuscript (please see page 8, line 22).

- "nor did we observe an effect of microbiota depletion (Fig. S1C)": I cannot find this result.

Response: We show in Fig. S1C (now Fig. S3C in the revised manuscript) that not only FFAR2/3 deficiency but also microbiota depletion did not influence production of most cytokines and chemokines.

- Fig. 4: "possibly partly due to the lower cell numbers analyzed". How many cell numbers where analyzed in each case?

Response: In total, 30,215 cells (including only 85 patrolling monocytes) have been analysed.

- Suppl. Fig. 4A. If the data is available, it will be nice to show that CSf2 KO mice have similar levels of IMs and PMNs and therefore the depletion is specific for AMs. A similar analysis could be done for Suppl. Fig 4B and Suppl. Fig. 4C.

Response: We have added data to show that lack of CSF2 does not affect PMNs and IMs, that depletion of PMNs does not influence numbers of AMs and IMs, and that depletion of IMs does not affect AM and PMN numbers (please see new Fig. S6A-I)

- Could you eliminate the last part of the statistical analysis since it seems to be instructions from the manuscript?

Response: This part has been deleted.

REVIEWERS' COMMENTS

Reviewer #1 (Remarks to the Author):

1. Thank you for addressing my concerns and providing additional data to support your conclusions. Given the constraints with available material and the demonstration of a change in circulating SCFAs, I'm happy with this response.
2. Congratulations on the new data and analysis. This is a significant addition to the study and helps strengthen the conclusions.
3. I appreciate the thoroughness of the cytokine analysis, and agree that this is a robust measure to support the conclusions.
- 4) I'm still unsure about the depth versus breadth of this analysis. I appreciate that they have used a large number of fecal donors - more than most studies do - but the depth of analysis of each human microbiota sample is therefore still limited. However, given the consistency of the resulting data and the conformation of transfer by sequencing, I'm not going to push for further work here.
5. Thank you for confirming that the neutrophil depletion is now statistically significant. This is an important addition to the study, and I'm are satisfied with the authors' responses.

Reviewer #3 (Remarks to the Author):

I would like to thank the authors for the effort done in replying to all my questions. They have provided all the information that I asked for, including new analysis and experiments that have increased the strength of their conclusions.

I do not have any more comments to do on the manuscript and just congratulate the authors for the nice work done.